# EWSR1-ATF1 dependent 3D connectivity regulates oncogenic and differentiation programs in Clear Cell Sarcoma

Emely Möller[1], Viviane Praz[1], Sanalkumar Rajendran[1], Rui Dong[2], Alexandra Cauderay[1,2], Yu-Hang Xing [2], Lukuo Lee[2], Carlo Fusco[1], Liliane C. Broye[1], Luisa Cironi[1], Sowmya Iyer[2], Shruthi Rengarajan[2], Mary E. Awad [2], Beverly Naigles [2], Igor Letovanec[3,4], Nicola Ormas[5], Giovanna Finzi[5], Stefano La Rosa[4,6], Fausto Sessa[6], Ivan Chebib[7], G. Petur Nielsen[7], Antonia Digklia[8], Dimitrios Spentzos [9], Gregory M. Cote[10], Edwin Choy[10], Martin Aryee [2,11], Ivan Stamenkovic[1], Gaylor Boulay[2,12], Miguel N. Rivera[2,11,12] & Nicolò Riggi [1,12✉]

Oncogenic fusion proteins generated by chromosomal translocations play major roles in cancer. Among them, fusions between EWSR1 and transcription factors generate oncogenes with powerful chromatin regulatory activities, capable of establishing complex gene expression programs in permissive precursor cells. Here we define the epigenetic and 3D connectivity landscape of Clear Cell Sarcoma, an aggressive cancer driven by the *EWSR1-ATF1* fusion gene. We find that EWSR1-ATF1 displays a distinct DNA binding pattern that requires the EWSR1 domain and promotes ATF1 retargeting to new distal sites, leading to chromatin activation and the establishment of a 3D network that controls oncogenic and differentiation signatures observed in primary CCS tumors. Conversely, EWSR1-ATF1 depletion results in a marked reconfiguration of 3D connectivity, including the emergence of regulatory circuits that promote neural crest-related developmental programs. Taken together, our study elucidates the epigenetic mechanisms utilized by EWSR1-ATF1 to establish regulatory networks in CCS, and points to precursor cells in the neural crest lineage as candidate cells of origin for these tumors.

[1] Experimental Pathology Service, Lausanne University Hospital and University of Lausanne, Lausanne, Switzerland. [2] Department of Pathology and Cancer Center, Massachusetts General Hospital, Charlestown, MA, USA. [3] Department of Histopathology, Central Institute, Valais Hospital, Sion, Switzerland. [4] Institute of Pathology, Lausanne University Hospital and University of Lausanne, Lausanne, Switzerland. [5] Department of Pathology, ASST Sette Laghi, Varese, Italy. [6] Pathology Unit, Department of Medicine and Surgery, University of Insubria, Varese, Italy. [7] Department of Pathology, Massachusetts General Hospital and Harvard Medical School, Boston, MA, USA. [8] Department of Oncology, Lausanne University Hospital and University of Lausanne, Lausanne, Switzerland. [9] Department of Orthopaedic Surgery, Massachusetts General Hospital, Boston, MA, USA. [10] Department of Medicine, Division of Hematology and Oncology, Massachusetts General Hospital, Boston, MA, USA. [11] Broad Institute, Cambridge, MA, USA. [12] These authors jointly supervised this work: Gaylor Boulay, Miguel N. Rivera, Nicolò Riggi. ✉email: Nicolo.Riggi@chuv.ch

Alterations in the mechanisms that regulate gene expression are key oncogenic events in many types of cancer. These events include mutations in transcription factors (TFs) and chromatin regulators, as well as changes in DNA methylation patterns, histone modifications, and higher order 3D chromatin structure[1]. EWSR1 fusion proteins are a class of oncogenes characterized by fusions of the prion-like disordered N-terminal domain of EWSR1 with a variety of TFs. The best characterized example is the EWSR1-FLI1 Ewing sarcoma fusion protein, where the presence of EWSR1 allows the fusion protein to bind genomic locations that are otherwise inaccessible to wild type (wt) FLI1. As a consequence, EWSR1-FLI1 can operate as a pioneer factor to activate de novo enhancers that shape the epigenetic landscape and identity of tumor cells[2,3]. Other EWSR1 fusion proteins, composed of the same EWSR1 domains fused to other TFs, remain less well understood but are expected to also function as aberrant transcriptional regulators.

Clear cell sarcoma (CCS) is a rare soft tissue tumor with high rates of recurrence and metastasis, and low 5- and 10-year overall survival rates (50 and 38%, respectively)[4]. CCS most frequently arises in the soft tissues of the extremities of young adults and surgical excision constitutes the mainstay of treatment, with some patients benefitting from additional radiotherapy[4,5]. CCS was previously termed malignant melanoma of soft parts due to its histological and molecular similarities with skin melanoma (MM), but the identification of the chromosomal translocation t(12;22)(q13;q12), or less frequently t(2;22)(q33;q12), has helped define this tumor as a separate entity[6,7]. Although CCS may display additional genomic aberrations, t(12;22)(q13;q12) is the only genetic alteration found in more than 90% of cases, and the resulting *EWSR1-ATF1* fusion gene is believed to represent the major driver of this disease (https://mitelmandatabase.isb-cgc.org). EWSR1-ATF1 expression alone is sufficient to induce CCS-like tumors in mice, and stem cells derived from the mesenchymal or neural crest lineages have been proposed to be particularly permissive for EWSR1-ATF1 dependent transformation[8–10].

Similar to other EWSR1 fusion proteins, *EWSR1-ATF1* encodes for an aberrant TF containing the transcriptional activation domain of EWSR1 and the DNA-binding and protein dimerization domains of ATF1. The activation properties of EWSR1-ATF1 depend on both EWSR1 and ATF1, and are stronger than that of wt ATF1 despite operating through the same cAMP-responsive elements (CRE motifs)[11,12]. ATF1 belongs to the ATF1/CREB/CREM subfamily of basic leucine zipper (bZIP) TFs that bind CRE motifs as homo- or hetero-dimers in their activated, phosphorylated state[13,14]. Chromatin immunoprecipitation (ChIP) studies similarly demonstrated direct binding of EWSR1-ATF1 to CREB/ATF1 motifs in promoters and/or enhancers of target genes, reinforcing the notion that the ATF1 domain confers DNA-binding specificity to the fusion protein[8,15,16].

Interestingly, two features distinguish EWSR1-ATF1 from other EWSR1 fusion proteins. First, EWSR1-ATF1 has been associated with multiple tumor types encompassing a wide histological spectrum and divergent in terms of clinical aggressiveness and prognosis. In addition to CCS, these entities include Angiomatoid Fibrous Histiocytoma (AFH), Myoepithelial tumor, Hyalinizing Clear Cell carcinoma, Clear Cell Odontogenic Carcinoma, Mesothelioma and Myxoid Mesenchymal tumors[17,18]. Second, the expression of wt ATF1 is retained in CCS, in contrast to several sarcomas driven by fusion proteins containing EWSR1, in which expression of the wt form of the EWSR1 fusion partner is lost[2,19]. Altogether these observations suggest that the oncogenic properties of EWSR1-ATF1 may be cell context dependent, and rely on the pre-existing landscape of the corresponding precursor cells. They also point to the potential involvement of wt

ATF1 in the pathogenesis of these tumors, and highlight our limited understanding of the molecular underpinnings driving these malignancies.

In this work, we define the genome-wide chromatin and DNA accessibility states associated with EWSR1-ATF1 binding sites in CCS cell lines, primary tumors, and mesenchymal stem cells (MSCs). In combination with 3D nuclear conformation and gene expression profiling, our study uncovers the gene regulation network associated with EWSR1-ATF1 and delineates the role of this fusion in establishing the proliferative and differentiation programs that characterize primary CCS tumors, and may underlie the biological similarities to MM. Conversely, we find that transcriptional programs activated upon EWSR1-ATF1 depletion are associated with the neural crest lineage in the human single-cell atlas and show connections to potential CCS cells of origin. Taken together our studies show how EWSR1-ATF1 reconfigures gene regulation networks in CCS to promote oncogenic programs and prevent normal differentiation pathways.

## Results

**EWSR1-ATF1 binding sites in CCS are distinct from endogenous wild type ATF1.** In order to study EWSR1-ATF1 function in CCS, we first identified EWSR1-ATF1 binding sites in two well-established CCS cell lines (DTC1 and SU-CCS-1). Given that wt ATF1 is expressed in CCS cells, we defined EWSR1-ATF1 binding sites by overlapping the results of ChIP sequencing (ChIP-seq) for the EWSR1 (N-terminus) with signals for ATF1 (C-terminus). This approach defined a set of 2385 EWSR1-ATF1 peaks (Supplementary Fig. 1a). The majority of EWSR1-ATF1 peaks (84%) were associated with distal regions and the remaining 16% of peaks were located at gene promoters (TSS) (Fig. 1a). In contrast, 60% of peaks bound by wt ATF1 (containing ATF1 signals but no EWSR1 signals), were located at promoters (Supplementary Fig. 1b). To further evaluate the difference in genomic distribution between EWSR1-ATF1 and wt ATF1, we examined ChIP-seq profiles of wt ATF1 generated by the ENCODE consortium in HepG2 and K562 cells. These profiles also showed that binding occurs predominantly at TSS regions (HepG2: 77%, K562: 70%, Supplementary Fig. 1b). Moreover, distal sites bound by EWSR1-ATF1 are less frequently observed in other cell types since they display lower average DNAse I signals in 113 human cell types profiled by ENCODE (GSE29692)[20], compared to wt ATF1 sites (Fig. 1b). These observations suggest that despite sharing the same DNA-binding domain, EWSR1-ATF1 binds a distinct set of enhancer elements. Noteworthy, the same EWSR1-ATF1 binding profile was also observed in a primary CCS tumor (Fig. 1c). These findings are reminiscent of the EWSR1-FLI1 fusion protein in Ewing sarcoma, where the prion-like domain of EWSR1 enables its binding to a set of de novo tumor specific sites[21], and indicate that EWSR1-ATF1 may also acquire similar epigenetic remodeling properties. The N-terminal portion of the FUS, TAF15 and EWSR1 proteins has been shown to interact directly with the SWI/SNF chromatin remodeling complex[22], suggesting that EWSR1-ATF1 may also be able to interact with this complex. To validate this hypothesis, we performed co-immunoprecipitation (co-IP) studies in DTC1 cells for the interaction between EWSR1-ATF1 / wt ATF1 and BRG1, a core SWI/SNF component. While the ATF1 C antibody was able to co-IP BRG1, together with both EWSR1-ATF1 and wt ATF1, the BRG1 antibody selectively pulled down EWSR1-ATF1 (Supplementary Fig. 1c), suggesting that the interaction with the SWI/SNF complex is a neo-morphic property of the fusion protein, which may, at least in part, explain its distinct chromatin binding profile.

To determine the chromatin states associated with EWSR1-ATF1 binding sites we next profiled the major histone

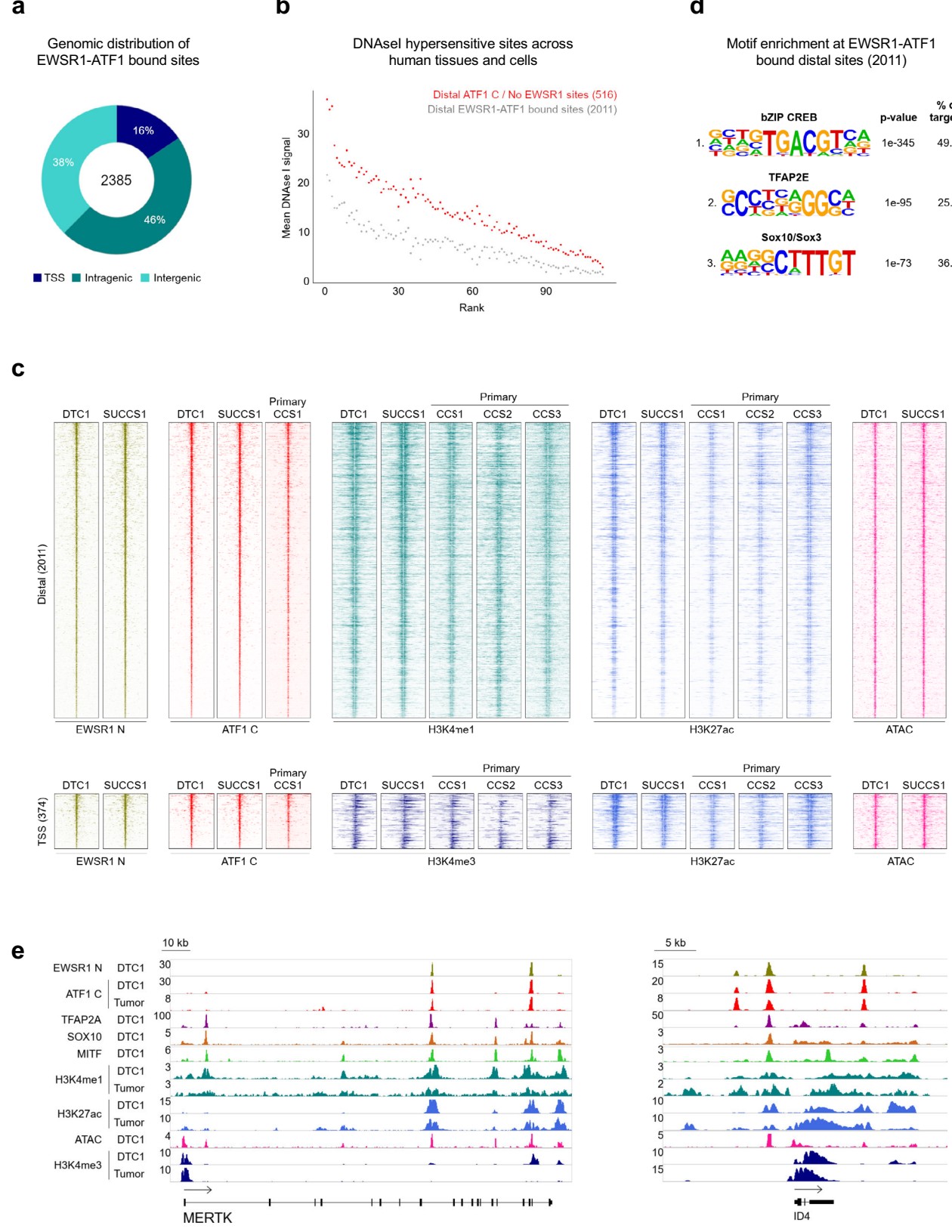

modifications H3K4me1, H3K27ac, and H3K4me3 in two replicates of DTC1 and SU-CCS-1, as well as in primary CCS. In addition, we used ATAC-seq to generate a map of chromatin accessibility in DTC1 and SU-CCS-1 cells. The majority of EWSR1-ATF1 distal binding sites were found to be associated with active enhancer elements, marked by the presence of

H3K4me1, H3K27ac, and ATAC signals[23,24] (Fig. 1c). Similarly, EWSR1-ATF1 binding sites at TSSs were associated with H3K4me3, H3K27ac, and ATAC signals (Fig. 1c), consistent with the notion that the fusion protein behaves as a transcriptional activator at both regulatory regions. Notably, the three primary tumors CCS1-CCS3 displayed similar chromatin state

**Fig. 1 EWSR1-ATF1 displays a distinct binding pattern compared to wt ATF1. a** Genomic distribution of 2385 consensus EWSR1-ATF1 binding sites shared between DTC1 and SU-CCS-1 cell lines, illustrating preferential EWSR1-ATF1 binding to distal genomic regions. **b** DNAse I hypersensitivity profile comparison between 2011 distal EWSR1-ATF1 sites and 516 wt ATF1 bound distal sites across 113 different cell types, showing a more restricted DNA accessibility pattern for the fusion protein binding sites in other cell types. **c** *Top panel*: heatmaps depicting EWSR1 N, ATF1 C, H3K4me1, H3K27ac, and ATAC signal intensities at 2011 EWSR1-ATF1 bound distal sites in DTC1 and SU-CCS-1 cell lines, as well as three primary CCS tumors. *Bottom panel*: EWSR1 N, ATF1 C, H3K4me3, H3K27ac, and ATAC signal intensities at 374 TSS-associated EWSR1-ATF1 binding sites. For each heatmap 20 kb regions centered on the EWSR1-ATF1 peaks are shown. Signals are ranked by ATF1 C intensity. **d** De novo motif enrichment analysis for 2011 distal EWSR1-ATF1 binding regions. The top three motifs identified are shown. Binomial p-values are given by the motif enrichment software HOMER. **e**, ChIP-seq track at MERTK and ID4 genomic loci, illustrating similar EWSR1-ATF1 binding and chromatin activity profiles between DTC1 and primary CCS tumor cells, and the presence of TFAP2A, SOX10, and MITF at EWR1-ATF1 sites.

profiles at EWSR1-ATF1 peaks (Fig. 1c). Given that EWSR1-ATF1 expression has been identified in additional tumor types, we performed similar chromatin profiling in two primary AFH tumors. AFH and CCS differ significantly at both histological and behavioral levels. AFH displays neither clear cell morphology nor a melanocytic differentiation phenotype and follows a relatively indolent clinical course with rare metastasis. AFH patients, therefore, have a markedly better prognosis than those diagnosed with CCS[25]. Interestingly, the AFH chromatin profiles at the 2385 EWSR1-ATF1 binding sites bore only partial similarity to those in primary CCS and CCS cell lines (Supplementary Fig. 1d *left*). A genome-wide survey of H3K4me1 and H3K27ac peaks revealed high-quality signals, excluding the possibility that these results might reflect lower ChIP-seq quality in AFH (Supplementary Fig. 1d *right*). Our observations, therefore, suggest that EWSR1-ATF1 action may be conditioned by properties intrinsic to the tumor and its cell of origin.

The most highly enriched motif at distal EWSR1-ATF1 bound sites was the consensus CRE motif followed by TFAP2 and SOX-related motifs (Fig. 1d). Because the CREB family of TFs recognizes CRE half-sites (TGACG/CGTCA) in the genome[26], the EWSR1-ATF1 peak regions that were not identified as having a full CRE site (TGACGTCA) were queried for their enrichment in short CRE motifs (4, 5, 6 nt). Indeed, the CRE half-site was identified as the top motif in those regions (Supplementary Fig. 1e). Our results are in line with previously published studies showing that the DNA-binding specificity of EWSR1-ATF1 is conferred by the ATF1 DNA-binding domain[8,11]. TFAP2A (aka AP-2α) and SOX10 play central roles in the biology of neural crest stem cells (NCSCs)[27] and enrichment for these motifs at EWSR1-ATF1 regulatory elements may reflect the proposed origin of CCS in undifferentiated NC-derivatives[8]. 52% of the EWSR1-ATF1 bound sites bearing the CRE motif, and 50% of those containing the CRE half-site, were co-enriched for TFAP2 and/or SOX motifs (Supplementary Fig. 1f), indicating that these TFs may bind those genomic sites in concert with the fusion protein. To investigate the potential cooperativity between the fusion protein and NCSC-related TFs, we profiled TFAP2A and SOX10 binding sites by ChIP-seq in both CCS cell lines, and observed the two TFs to be present at the majority of the 2385 EWSR1-ATF1 bound sites (TFAP2A: 59% in DTC1 and 82% in SU-CCS-1, SOX10: 56% in DTC1 and 83% in SU-CCS-1, Supplementary Fig. 1g). In addition, given the known functional collaboration between TFAP2A / SOX10 and MITF to regulate melanocytic differentiation[28] and considering the pivotal role of this TF during CCS development[15], we also generated DNA binding profiles for MITF in both cell lines. Similar to TFAP2A and SOX10, MITF showed marked co-occupancy at EWSR1-ATF1 bound sites (73% in DTC1 and 89% in SU-CCS-1, Supplementary Fig. 1g). Notably, enhancers bound by EWSR1-ATF1 as well as TFAP2A, SOX10, and MITF were found proximal to genes such as *MERTK* and *ID4* (Fig. 1e) that have previously been shown to be selectively expressed in CCS

compared to other soft tissue sarcomas and melanoma[29], as well as to many novel targets. Together, these results suggest that the fusion protein may hijack the pre-existing TF network of tumor precursor cells to establish its oncogenic program.

**EWSR1-ATF1 knockdown shows marked effects on chromatin activation states.** Next, we sought to determine the functional impact of EWSR1-ATF1 on chromatin activity. To this end, we used a siRNA-based strategy to deplete the fusion transcript in both DTC1 and SU-CCS-1 cell lines. The selected siRNA specifically targets the *EWSR1-ATF1* breakpoint sequence without altering the expression of wt *ATF1* (Supplementary Fig. 2a), as observed in an earlier study[10]. Optimal knock-down (KD) efficiency was observed at 96 h after transfection (Fig. 2a), providing the time-point for all subsequent analyses. The chromatin profiles of EWSR1-ATF1-depleted cells showed a marked reduction in ATF1 C signal across all binding sites (Fig. 2b, c *left*), validating the relevance of our EWSR1-ATF1 reference peak set. Overall, the two cell lines responded similarly to EWSR1-ATF1 depletion with a significant decrease in enhancer activity, as illustrated by the reduction in both H3K4me1 and H3K27ac signals at distal binding sites (Fig. 2c *middle*), as well as a reduction in H3K4me3 and H3K27ac signals at TSS sites (Supplementary Fig. 2b). Importantly, the reduction in chromatin activation was associated with a significant decrease in DNA accessibility at the same genomic regions, as assessed by ATAC-seq (Fig. 2c *right* and Supplementary Fig. 2b). Accordingly, we observed reduced activity and accessibility of enhancers proximal to the CCS-specific genes *MERTK* and *ID4* (Fig. 2d). In contrast, genome-wide H3K27ac signals showed both decreases and increases in EWSR1-ATF1-depleted DTC1 and SU-CCS-1 tumor cells (Supplementary Fig. 2c), attesting to profound changes in their global epigenetic landscape. These results confirm the ability of EWSR1-ATF1 to induce chromatin activation at its binding sites, and further suggest a role for the fusion protein in maintaining both the basal (H3K4me1) and activated (H3K27ac) enhancer states. The observed decrease in the ATAC signal also suggests that EWSR1-ATF1 may be involved in generating de novo distal regulatory elements through pioneering properties similar to the ones identified for EWSR1-FLI1 in EwS[2].

The residual chromatin activation observed after EWSR1-ATF1 depletion may be due to the short time frame of our experiments or may potentially be ascribed to cooperating TFs that fine-tune EWSR1-ATF1 activity. To investigate the co-dependency between EWSR1-ATF1 and its collaborative TF network, we generated genome-wide binding profiles of TFAP2A, SOX10, and MITF in EWSR1-ATF1-depleted SU-CCS-1 cells. Consistent with the notion that the fusion protein recruits these TFs and hijacks the transcriptional network of tumor precursor cells, all three TFs were significantly displaced from the EWSR1-ATF1 binding sites upon removal of the fusion protein (Supplementary Fig. 2d). In aggregate, these results define a network of TFs associated with EWSR1-ATF1, and whose

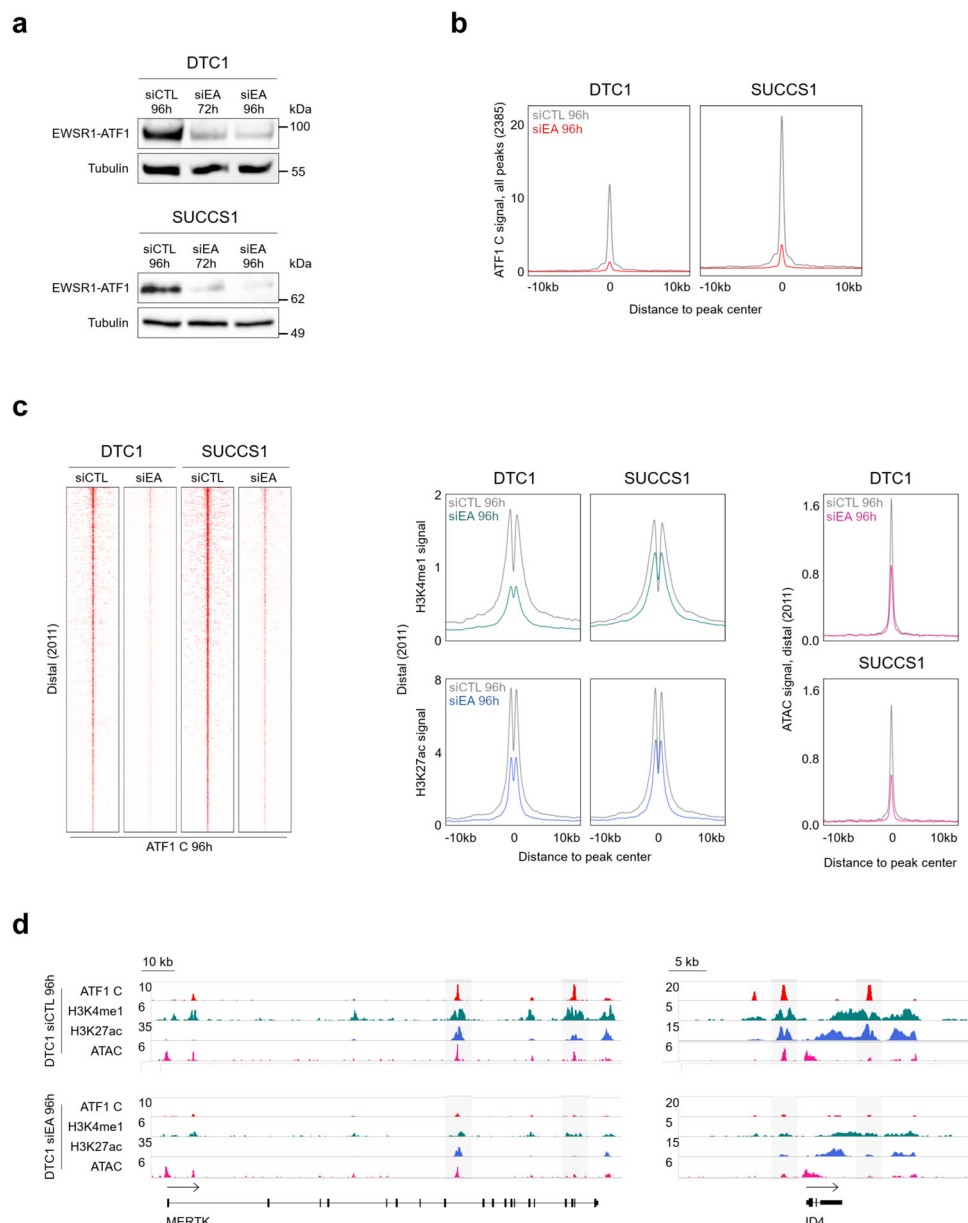

**Fig. 2 EWSR1-ATF1 depletion reduces chromatin activation and DNA accessibility. a** Western blot analysis of EWSR1-ATF1 protein levels in DTC1 and SU-CCS-1 cell lines transfected with a siRNA targeting the fusion gene (siEA 72 h and 96 h) or an unrelated sequence (siCTL 96 h). Tubulin protein levels were used as loading control. The experiment was independently repeated twice with similar results. Source images are provided in the Source Data file. **b** Composite plots showing the marked reduction in ATF1 C ChIP-seq signal at 2385 EWSR1-ATF1 bound sites in DTC1 and SU-CCS-1 cells transfected with either siEA or siCTL siRNAs for 96 h. **c** *Left panel*: Heatmaps depicting the distribution of ATF1 C ChIP-seq signal across 2011 EWSR1-ATF1 bound distal sites in siEA or siCTL-treated DTC1 and SU-CCS-1 cells at 96 h. Heatmaps show 20 kb genomic regions centered on EWSR1-ATF1 peaks ranked by ATF1 C signal intensity. *Middle panel*: composite plots of H3K4me1 (*upper panels*) and H3K27ac (*lower panel*s) signal at 2011 EWSR1-ATF1 bound distal sites in siEA or siCTL-treated DTC1 and SU-CCS-1 cells at 96 h, showing a reduction in both enhancer marks upon EWSR1-ATF1 depletion. *Right panel*: composite plots of ATAC signal in siEA or siCTL-transfected DTC1 and SU-CCS-1 cells at 96 h, showing the reduction in chromatin accessibility associated with the loss of the fusion protein. **d** ChIP-seq and ATAC-seq tracks at the MERTK and ID4 genomic loci (as in Fig. 1e) in siEA or siCTL-treated DTC1 cells at 96 h, showing the decrease in chromatin activity and accessibility induced by the loss of EWSR1-ATF1. EWSR1-ATF1 binding regions are highlighted in gray.

chromatin occupancy partially depends on the presence of the fusion protein.

**Wild type ATF1 is present at a subset of EWSR1-ATF1 binding sites**. The transition from a closed to a fully active chromatin state requires the interplay between cooperative TFs and chromatin remodeling complexes and co-activators[24]. Based on the role of wt ATF1 as an activator in other cellular models[13,30], we reasoned

that maintenance of its expression in CCS, coupled with enrichment in the ATF1 motif at EWSR1-ATF1 binding sites, may argue in favor of a cooperation between EWSR1-ATF1 and wt ATF1 in chromatin activation. To test this hypothesis, we assessed the degree of wt ATF1 occupancy at EWSR1-ATF1 binding sites by ChIP-seq in both DTC1 and SU-CCS-1 cell lines, as well as in one primary CCS, using an antibody specifically targeting the N-terminal portion of ATF1 (ATF1 N) which is lost in the fusion protein. Wt ATF1 was found to be present at the

majority of the EWSR1-ATF1 binding sites in both cell lines (62.5% in DTC1, 75% in SU-CCS-1, Fig. 3a), and confirmed in the primary CCS1 tumor, although at a lower percentage of EWSR1-ATF1 sites (38%, Fig. 3a), a difference that may be related to the technical challenge of profiling TFs in frozen tumor samples. Interestingly, when wt ATF1 occupancy was profiled in EWSR1-ATF1 KD cells, we observed a marked decrease in the ATF1 N signal at the majority of EWSR1-ATF1 binding sites (Fig. 3b). Notably, this reduction was selectively observed at binding sites of the fusion protein, as we did not observe significant changes in ATF1 N signal at genomic regions that were not co-bound by EWSR1-ATF1 (defined as regions showing overlapping ATF1 C and ATF1 N, but lacking EWSR1 N signal) (Fig. 3b).

These observations suggest that wt ATF1 occupancy at EWSR1-ATF1 binding sites depends on the presence of the fusion protein itself, and may at least in part explain the retargeting of wt ATF1 to distal genomic regions observed in CCS cells. To validate the physical interaction between EWSR1-ATF1 and wt ATF1, we transfected HEK293T cells with an expression construct encoding C-terminally V5-tagged EWSR1-ATF1 (EA-CV5) and performed co-IP using an anti-V5 tag antibody. This approach confirmed the interaction between EWSR1-ATF1 and endogenous wt ATF1, which was co-immunoprecipitated in EA-CV5-expressing but not empty vector-infected 293 T cells (CTL) (Fig. 3c *left*). We also confirmed the direct interaction between the endogenous EWSR1-ATF1 and wt ATF1 proteins in CCS cells by performing co-IP using either anti-ATF1 C or -ATF1 N antibodies, showing that the wt ATF1-specific anti-ATF1 N antibody pulled down EWSR1-ATF1 (Fig. 3c *right*). Our results support the notion that EWSR1-ATF1 recruits wt ATF1 to its binding sites. As EWSR1-ATF1 binding sites are more distally located than those of wt ATF1 (Supplementary Fig. 1b), this interaction leads to a genomic redistribution of wt ATF1.

**EWSR1-ATF1 recruits wild type ATF1 for chromatin activation**. We next considered whether the recruitment of wt ATF1 may support EWSR1-ATF1 function at its binding sites. To this end, we interrogated our ChIP-seq data generated in DTC1 and SU-CCS-1 cells for the association between wt ATF1 occupancy and H3K27ac signal intensity at EWSR1-ATF1 distal sites by calculating the correlation between the EWSR1 N, ATF1 N, and H3K27ac ChIP-seq scores. EWSR1 N and ATF1 N scores showed a significant correlation ($r = 0.48$, $p$-value 7.4e−115) (Fig. 3d), supporting the co-localization of EWSR1-ATF1 and wt ATF1 at those binding sites. Interestingly, H3K27ac scores correlated better with ATF1 N ($r = 0.42$, $p$-value 3.54e-88) than with EWSR1 N ($r = 0.08$, $p$-value 0.0002) (Fig. 3d), suggesting that wt ATF1 occupancy may constitute an important factor in promoting enhancer activity. To obtain deeper insight, we performed p300 ChIP-seq profiling in the CCS cell lines and found that p300 scores correlated better with ATF1 N ($r = 0.71$, $p$-value 3.51e−286) than with EWSR1 N ($r = 0.54$, $p$-value 1.67e−137) at EWSR1-ATF1 distal sites (Fig. 3e). These results suggest that the recruitment of wt ATF1 by EWSR1-ATF1 may participate in the process of chromatin activation by enhancing p300 recruitment and H3K27ac deposition at the fusion protein binding sites. To validate this hypothesis, we selected a lentiviral shRNA targeting specifically the 5′ end sequence of the wt *ATF1* transcript, not included in the *EWSR1-ATF1* transcript, and depleted wt ATF1 from DTC1 and SU-CCS-1 cells. A first attempt to achieve a complete knock down resulted in massive cell apoptosis, preventing any further downstream analysis. To circumvent this problem, we, therefore, decided to fine-tune the lentiviral load and experimental timing to obtain a partial but significant

depletion of wt ATF1 at 72 h post-lentiviral infection, without affecting the expression of the fusion protein (Fig. 3f). Despite the moderate decrease in ATF1 N signal at the EWSR1-ATF1 binding sites, we observed a concomitant reduction in H3K27ac signal (Supplementary Fig. 3), and in the expression level of a panel of genes whose regulatory elements show EWSR1-ATF1 and wt ATF1 co-occupancy in both CCS lines (Fig. 3g), supporting the notion that wt ATF1 participates in the transcriptional activity of EWSR1-ATF1.

To further address the functional contribution of EWSR1-ATF1 and its interplay with wt ATF1 we turned to primary pediatric mesenchymal stem cells (hpMSCs), a putative precursor model for a subset of human sarcomas. Importantly, the permissiveness of MSCs toward EWSR1-ATF1-driven transformation has been previously demonstrated in a transgenic CCS murine model[9]. We introduced C-terminally tagged EWSR1-ATF1 (EA-CV5) into hpMSCs (Supplementary Fig. 4a) and profiled these cells by ChIP-seq with anti-V5 tag, -ATF1 N, -H3K27ac, -H3K4me3 and p300 antibodies, as well as by ATAC-seq. V5 ChIP-seq identified 60% of the EWSR1-ATF1 binding sites observed in CCS cells (Fig. 4a), which showed a similar bias toward binding distal locations (82% distal and 18% TSS-associated) (Supplementary Fig. 4a).

First, we investigated the presence of wt ATF1 at EWSR1-ATF1 binding sites in EA-CV5 hpMSCs shared with CCS. We divided the peaks into 3 categories: peaks uniquely occupied by EWSR1-ATF1 (No ATF1, $n = 449$) or peaks co-occupied by both the fusion protein and wt ATF1, defined as either De Novo if lacking ATF1 peaks in control cells ($n = 398$), or Pre-Existing if ATF1 was present ($n = 475$). No ATF1 and De Novo sites were to a larger extent located at distal genomic regions (88%, $n = 395$, and 85%, $n = 340$, respectively), as compared to Pre-existing sites (77%, $n = 364$) (Supplementary Fig. 4b), and displayed increased ATAC signal in EA-CV5 cells (Fig. 4b). These two categories, however, differed in H3K27ac activation mark signals. While De Novo sites showed a clear increase in H3K27ac signal paired with p300 recruitment, No ATF1 sites lacked accumulation of both markers (Fig. 4b). The pre-existing sites were also characterized by an increase in ATAC, H3K27ac, and p300 signals in EA-CV5 cells (Supplementary Fig. 4c). However, the increase in H3K27ac signal was less pronounced, probably due to variable pre-existing H3K27ac levels that may modulate their responsiveness to EWSR1-ATF1-mediated activation. Similar results for No ATF1, De Novo, and Pre-existing peak categories were obtained with N-terminal V5-tagged EWSR1-ATF1 (NV5-EA) (Supplementary Fig. 4d). Taken together, these results indicate that to achieve complete chromatin activation, EWSR1-ATF1 requires wt ATF1 recruitment and cooperativity at its direct binding sites. A similar pattern was observed by calculating ChIP score correlations between EWSR1-ATF1, ATF1, and H3K27ac. Similar to our analyses in CCS cell lines (Fig. 3d), H3K27ac showed a stronger correlation with ATF1 N ($r = 0.59$, p-value 6.44e-112) than with V5 ($r = 0.24$, $p$-value 1.86e−17) (Fig. 4c), suggesting that H3K27ac deposition at EWSR1-ATF1 binding sites may indeed require wt ATF1 cooperativity.

**EWSR1-ATF1 binding at distal sites is dependent on the EWSR1 prion-like domain**. Interrogation of the three subsets of EWSR1-ATF1 sites for DNAse I signals in the same ENCODE panel of 113 distinct cell types used in Fig. 1b revealed that No ATF1 and De Novo sites had a markedly lower DNAse I sensitivity signal in other cell types compared to the Pre-existing sites (Fig. 4d). In contrast, the Pre-existing sites that are pre-bound by wt ATF1 displayed a strong DNAse I signal across a majority of other cell types, suggestive of genomic regions more widely

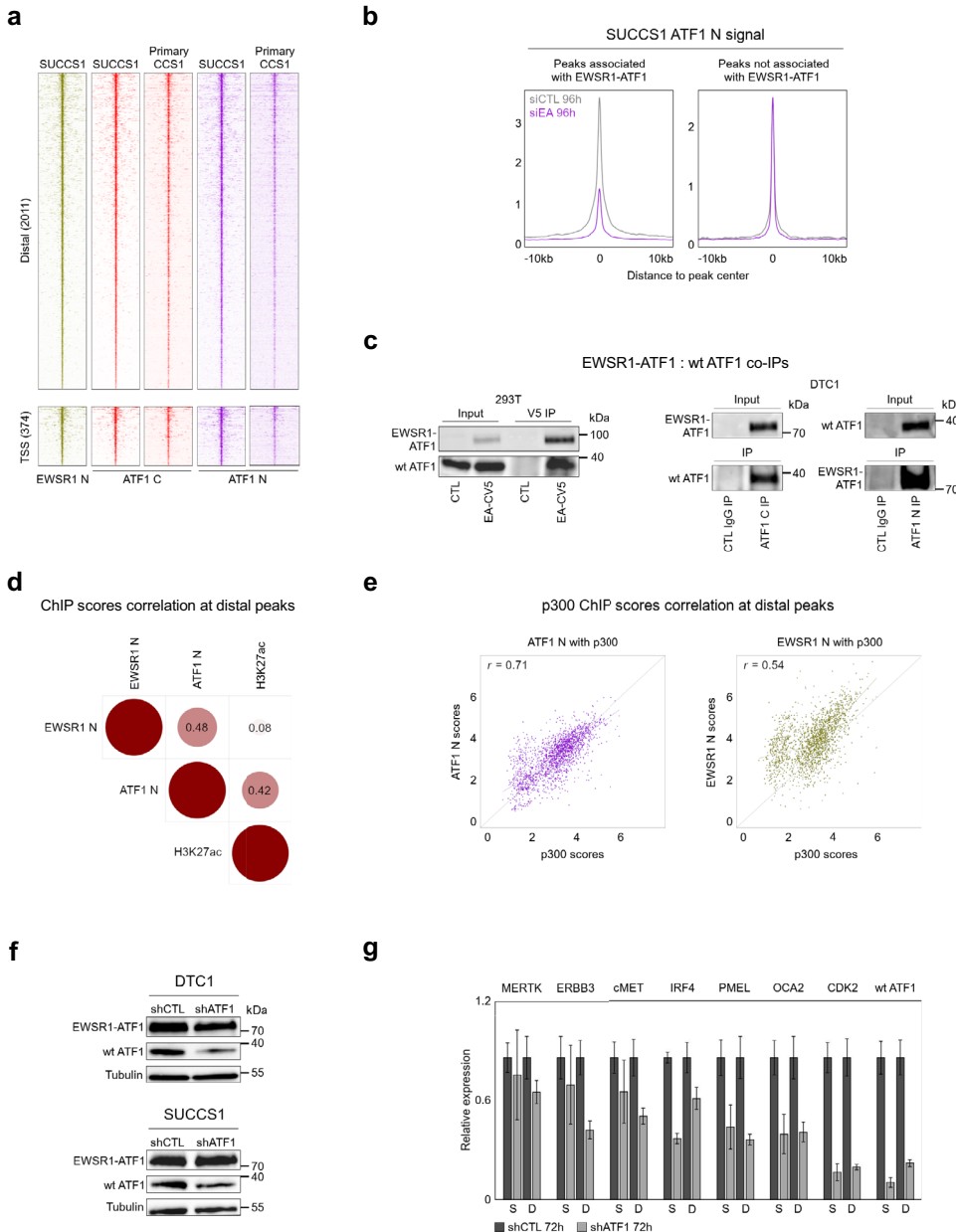

**Fig. 3 wt ATF1 is present at a majority of EWSR1-ATF1 binding sites. a** Heatmaps depicting EWSR1 N, ATF1 C and ATF1 N ChIP-seq signal intensities at 2011 distal and 374 proximal EWSR1-ATF1 binding sites in SU-CCS-1 and primary CCS1 tumor cells, illustrating genomic co-occupancy by the wt ATF1 TF at the fusion protein binding sites. For each heatmap 20 kb regions centered on the EWSR1-ATF1 peaks are shown. Signals are ranked by ATF1 C intensity. **b** Composite plots showing changes in ChIP-seq signal for ATF1 N peaks associated (*left*) or not (*right*) with EWSR1-ATF1 binding in SU-CCS-1 cells transfected with a siRNA targeting the fusion gene (siEA) or control (siCTL) siRNAs. The loss of the fusion protein is associated with a decrease in ATF1 N signal only at EWSR1-ATF1 bound sites, suggesting wt ATF1 recruitment by the fusion protein. **c** Co-immunoprecipitation (Co-IP) assay showing the direct interaction between the EWSR1-ATF1 and wt ATF1 proteins in 293 T (*left*) and DTC1 cells (*right*). 293 T cells were transfected with either a V5-tagged EWSR1-ATF1 (EA-CV5) or empty vector (CTL) constructs. The IP was performed using an anti-V5 tag antibody, and the western blot revealed using anti-V5 (*top*) or anti-ATF1 C (*bottom*) antibodies. DTC1 IPs were performed using anti-ATF1 C or -ATF1 N antibodies, compared to control (CTL) IgG, and the western blot revealed using an anti-ATF1 C antibody. The experiment was independently repeated twice with similar results. **d** Correlation plot of EWSR1 N, ATF1 N, and H3K27ac ChIP-seq scores at the 2011 distal regions, showing stronger correlation between H3K27ac deposition and wt ATF1 presence, than with EWSR1-ATF1 presence (represented by ATF1 N and EWSR1 N signals, respectively). **e** Scatter plots of ATF1 N and p300 (*left*), or EWSR1 N and p300 (*right*) ChIP-seq scores, illustrating that wt ATF1 presence correlates better with p300 occupancy than EWSR1-ATF1. **f** Western blot analysis of EWSR1-ATF1 and wt ATF1 protein levels in DTC1 and SU-CCS-1 cell lines infected with a shRNA targeting the wt ATF1 transcript (shATF1) or an unrelated sequence (shCTL) at 72 h post-lentiviral infection. Tubulin protein levels were used as loading control. The experiment was independently repeated twice with similar results. **g** qPCR analysis showing reduction of *wt ATF1* transcripts, as well as a panel of EWSR1-ATF1 / wt ATF1 co-regulated target transcripts, in shATF1- vs shCTL-infected SU-CCS-1 (S) and DTC1 (D) cells at 72 h post-lentiviral infection. Three replicates of each condition were used to calculate mean Ct values, mean relative expression values were then calculated according to the $2^{-\Delta\Delta Ct}$ method relative to the shCTL sample values, and normalized to the endogenous control gene *GAPDH*. Error bars show standard deviation of mean values ($n = 3$ sample replicates). The experiment was independently repeated twice with similar results. Source data are provided in the Source Data file.

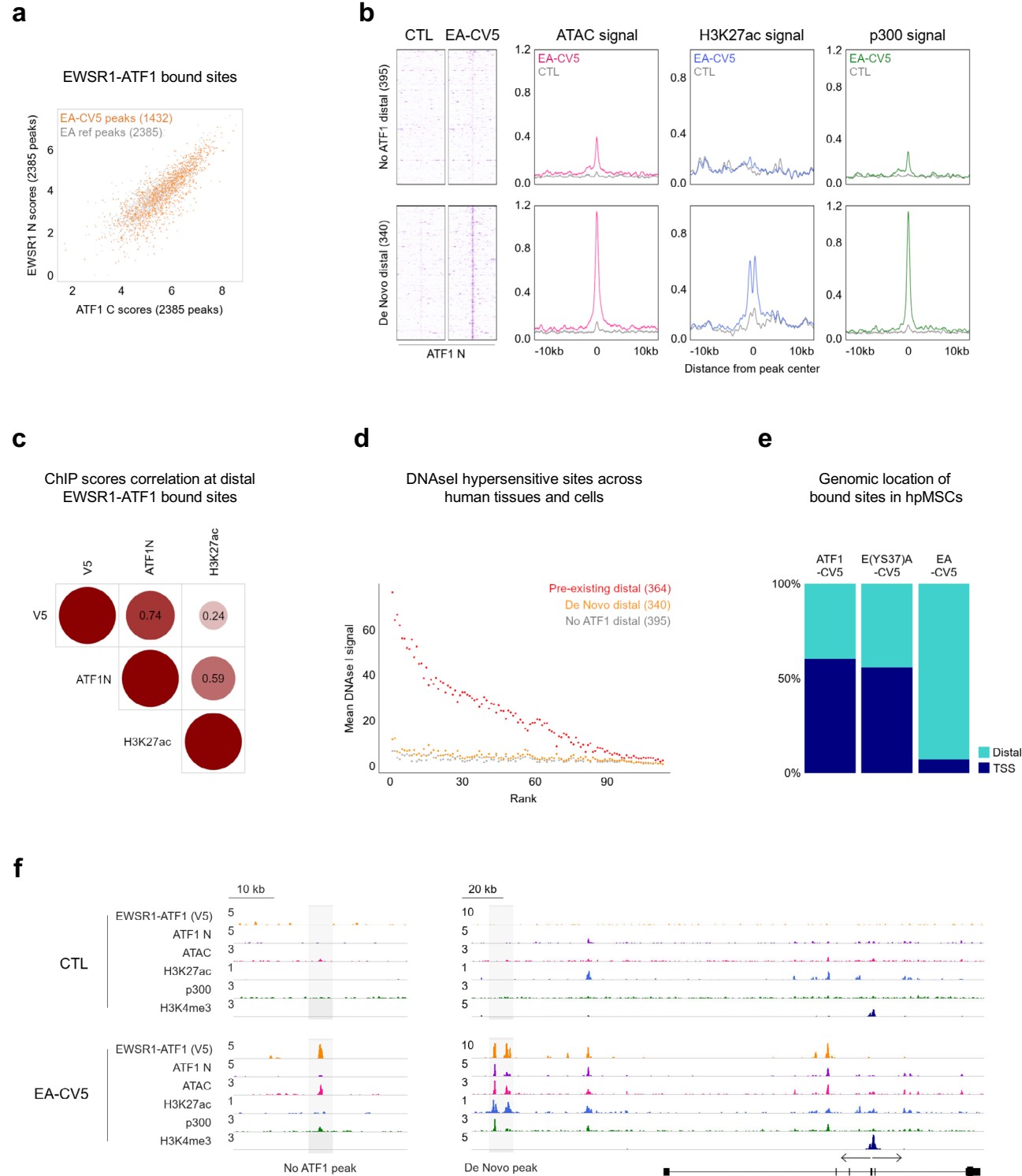

accessible to this TF. Given the ubiquitous expression of wt ATF1 and its genomic redistribution by the fusion protein in CCS, these results may reflect the ability of EWSR1-ATF1 to recruit wt ATF1 to genomic regions that would remain partially inaccessible in the absence of the fusion protein. They also raise the question of potential neo-morphic properties provided by EWSR1 to EWSR1-ATF1 function, including the capability to interact with the SWI/SNF remodeling complex (Supplementary Fig. 1c).

We previously demonstrated that the prion-like disordered domain of EWSR1 enables the EWSR1-FLI1 Ewing sarcoma

fusion protein to activate genomic locations that are inaccessible to wt FLI1[21]. These properties are abolished by mutations that alter the phase transition properties of EWSR1 (EWSR1-FLI1 YS37 mutant). To investigate whether a similar mechanism may underlie the genomic retargeting of wt ATF1 to distal sites, we generated a EWSR1-ATF1 YS37 mutant and expressed it in hpMSCs, along with EWSR1-ATF1 and wt ATF1 (Supplementary Fig. 4e). Importantly, the EWSR1-ATF1 YS37 mutant was found to retain the ability to interact with wt ATF1, (Supplementary Fig. 4f), despite losing its beta-isox-induced precipitation

**Fig. 4 EWSR1-ATF1 distal binding and wt ATF1 recruitment depend on EWSR1 prion-like domain. a** Scatter plot of EWSR1 N and ATF1 C ChIP-seq scores for the 1432 EWSR1-ATF1-V5 peaks (EA-CV5) shared between hpMSCs and CCS cell lines, showing an even distribution of V5 signal across all sites (EA ref peaks). **b** Heatmaps (*left*) depicting ATF1 N signal intensity at 395 and 340 No ATF1 and De Novo, respectively, EWSR1-ATF1-V5 bound distal sites in hpMSCs (EA-CV5) vs control (CTL) cells, illustrating the de novo recruitment of wt ATF1. 20 kb regions centered on the EWSR1-ATF1 peaks are shown. Composite plots depicting ATAC, H3K27ac, and p300 signal intensity at No ATF1 and De Novo distal sites, showing that the wt ATF1 recruitment by the fusion protein enhances its chromatin activation properties. **c** Correlation plot of V5, ATF1 N, and H3K27ac ChIP-seq scores at the 1174 distal EWSR1-ATF1-V5 bound sites in hpMSCs, confirming that H3K27ac deposition correlates better with wt ATF1 (ATF1 N) presence than with EWSR1-ATF1 (V5) presence, at the fusion protein binding sites observed in CCS cell lines. **d** DNAse I hypersensitivity profile comparison between No ATF1, De Novo, and Pre-existing hpMSCs EWSR1-ATF1-V5 distal bound sites across 113 different cell types, showing the restricted DNA accessibility pattern of genomic regions initially devoid of wt ATF1 occupancy. **e** Bar plots depicting the genomic distribution of EWSR1-ATF1 (EA-CV5), EWSR1(YS37)-ATF1 (E(YS37)A-CV5), and wt ATF1 (ATF1-CV5) binding sites in MSCs, as assessed by ChIP-seq using an anti-V5 antibody. **f** ChIP-seq, and ATAC-seq tracks showing distinct patterns of wt ATF1 recruitment, DNA accessibility, and chromatin activity changes at No ATF1 (*left*) and De Novo (*right*) EWSR1-ATF1-V5 binding sites in hpMSCs (highlighted in grey).

properties (Supplementary Fig. 4g). To interrogate the genomic distribution of the three TFs, hpMSCs were profiled by ChIP-seq using an anti-V5 antibody, because all three constructs included a V5 tag at their C-terminus. Consistent with our previous results, the binding pattern of EWSR1-ATF1 remained predominantly restricted to distal genomic regions (7% of TSSs), whereas the YS37 mutant displayed a significant increase in promoter binding (56% of TSS), close to the binding pattern of wt ATF1 (60% of TSS) (Fig. 4e).

In summary, our results show that EWSR1-ATF1 can bind inactive sites to initiate chromatin activation de novo upon wt ATF1 recruitment (Fig. 4f), whereas at active genomic regions pre-marked by wt ATF1, EWSR1-ATF1 leads to moderate increases in both chromatin activity and wt ATF1 occupancy. Our hpMSCs results also illustrate how the preferential binding pattern of EWSR1-ATF1 to distal sites identified in CCS cell lines relies on phase transition properties, and strongly contributes to wt ATF1 retargeting.

**Identification of the 3D connectivity of EWSR1-ATF1 binding sites in CCS.** Given that the vast majority of EWSR1-ATF1 binding sites are located at distal genomic regions, we next sought to better define the 3D connectivity of the fusion protein by performing 3D chromatin conformation profiling of DTC1 and SU-CCS-1 tumor cells by H3K27ac Hi-ChIP. This technology allows to determine the looping pattern associated with the activation mark H3K27ac and therefore provides means of identifying the 3D interactions of active distal regulatory sites[31]. We identified a total of 79497 and 41325 chromatin loops in SU-CCS-1 and DTC1 cells, respectively (Fig. 5a, b, and Supplementary Fig. 5a), of which 48 and 32% were associated with EWSR1-ATF1. These observations highlight the importance of the fusion protein in shaping both chromatin activity and conformation, since EWSR1-ATF1 binding sites represent a minority of H3K27ac peaks in both cell lines but are part of a large fraction of the chromatin loops detected by Hi-ChIP (Fig. 5c and Supplementary Fig. 5b). Notably, the fusion protein binding sites were associated with a higher number of loops compared to other H3K27ac peaks, and these loops were longer and had higher read counts (Fig. 5d and Supplementary Fig. 5c, d).

We then used Hi-ChIP profiles to define the direct target genes of each EWSR1-ATF1 binding site in both cell lines, and to generate a robust set of 2014 direct target genes shared between SU-CCS-1 and DTC1 cells (Fig. 5e, f, Supplementary Data 1). Out of the 2014 direct target genes identified by DTC1 and SU-CCS-1 Hi-ChIP, 535 were differentially expressed in siEA cells (Fig. 6a), and the majority ($n = 417$) decreased in expression upon EWSR1-ATF1 depletion. Remarkably, functional analysis of the 417 downregulated genes revealed their strong involvement in cell cycle regulation, consistent with the pro-oncogenic activity of the

fusion protein (Fig. 6b *left*, Supplementary Data 1). This gene expression program was also activated in our hpMSCs model, where a majority of the direct targets down-regulated in siEA cells showed induction upon EWSR1-ATF1 expression (Fig. 6b *right*). In addition, when comparing the transcriptional profiles of CCS and AFH primary tumors, we observed our down-regulated target gene repertoire to be more highly expressed in CCS samples (Fig. 6b, *right*), despite the presence of the same fusion gene in both tumor types.

We next sought to determine the changes in connectivity induced by the loss of the fusion protein in CCS tumor cells. For this purpose, SU-CCS-1 cells transfected with either siCTL or siEA were profiled by H3K27ac Hi-ChIP to identify differences in looping patterns. After normalization, we found 25008 loops with decreased and 19107 with increased counts (2-fold threshold) in SU-CCS-1 cells depleted of EWSR1-ATF1 (Fig. 6c). Decreases in loop counts were mostly observed at EWSR1-ATF1 associated loops (Fig. 6d, e), supporting the role of the fusion protein in chromatin activation. In contrast, the majority of increases in chromatin interactions were associated with genomic regions devoid of EWSR1-ATF1 binding sites (Fig. 6d, *top*).

**Coordinated changes in 3D connectivity and chromatin activity point to candidate CCS cell of origin.** Our group and others have previously shown that EWSR1-FLI1 depletion in EwS leads to the emergence of expression programs associated with MSCs, the putative cells of origin of this tumor[32]. We thus reasoned that new 3D connections identified in EWSR1-ATF1-depleted cells may be driven by regulatory elements that control transcriptional programs associated with candidate CCS cells of origin. To evaluate this hypothesis, H3K27ac profiles for siCTL and siEA-transfected SU-CCS-1 cells were surveyed for "de novo" H3K27ac regions, defined as peaks showing less than 5 reads in siCTL cells, and increasing at least four-fold in siEA cells. This analysis yielded 4063 and 1465 "de novo" distal and proximal sites, respectively, in EWSR1-ATF1-depleted SU-CCS-1 cells (Fig. 7a). The genomic coordinates of these sites were then used to interrogate matched H3K27ac Hi-ChIP profiles and define their direct target genes (Fig. 7b). Using adjusted p-value 0.05 and FC 1.5 cutoffs we identified 157 genes associated with "de novo" H3K27ac sites to be induced upon EWSR1-ATF1 removal in both DTC1 and SU-CCS-1 cells (Fig. 7c). Surprisingly, among the induced transcripts we observed TFs related to the ones that co-localize with EWSR1-ATF1 at its direct binding sites, including TFAP2A, SOX10, and MITF (Fig. 7d and Supplementary Data 1). In addition, when surveying global RNA-seq profiles of siEA-transfected DTC1 and SU-CCS-1 cells we also found induction of genes involved in melanosome biogenesis and melanin pigmentation, including MLANA, TYRP1, and PMEL which are associated with terminally differentiated melanocytes (Fig. 7d and

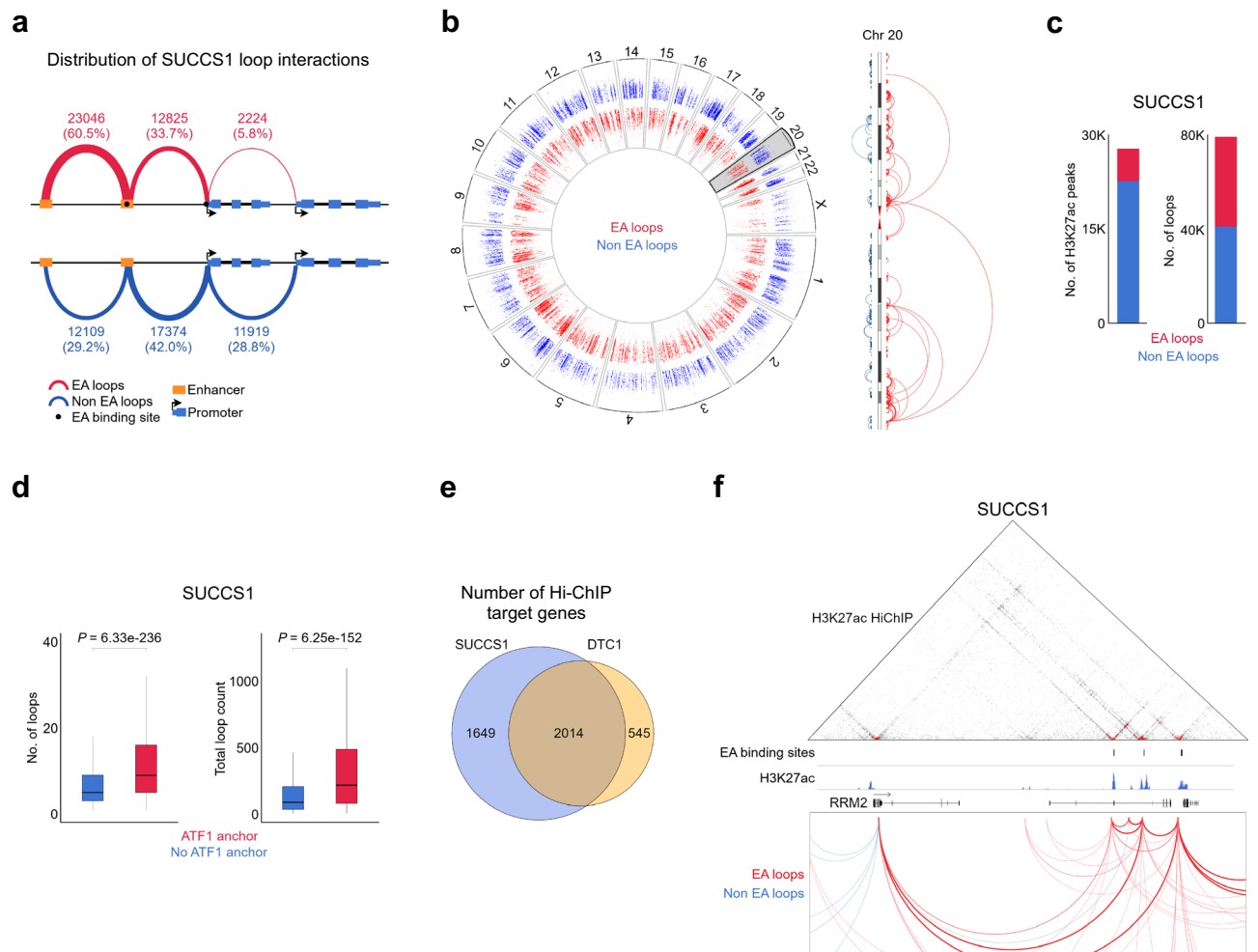

**Fig. 5 EWSR1-ATF1-associated connectivity dominates the 3D landscape of CCS tumor cells. a** Three-dimensional connectivity of EWSR1-ATF1-associated (EA loops, red) or –independent (Non EA loops, blue) chromatin loops in SU-CCS-1 tumor cells, showing a more distal-to-distal interaction pattern for fusion protein-connected loops. **b** *Left*: circos plot depicting all EWSR1-ATF1-associated or -independent chromatin loops genome-wide. The height of the blue and red signals inside the circos corresponds to the loop's length depicted in log10 scale. *Right*: higher magnification of chromosome 20 illustrating increased length for EWSR1-ATF1-associated loops, as compared to all other loops. **c** Bar plots showing the association of H3K27ac peaks (*left*) and chromatin loops (*right*) with EA loops (5121 and 38095, respectively) or Non-EA loops (22654 and 41402, respectively). **d** Box plots depicting the median number (*left*) and intensity (*right*) of chromatin loops associated (n = 38095) or not (n = 41402) with EWSR1-ATF1 binding sites. The lower and upper hinges correspond to the first and third quartiles (the 25th and 75th percentiles). The upper whisker extends from the hinge to the largest value no further than 1.5 * IQR from the hinge (where IQR is the inter-quartile range). The lower whisker extends from the hinge to the smallest value at most 1.5 * IQR of the hinge. Statistical significance was calculated by two-sided t-test (n = 4181 forATF1 anchors, n = 16756 for No ATF1 anchors). **e** Venn diagram presenting the number of direct target genes identified by Hi-ChIP as connected to the 2385 EWSR1-ATF1 binding sites in SU-CCS-1 and DTC1 tumor cells. **f** Hi-ChIP interaction map (*top*) and chromatin looping profile (*bottom*) for the RRM2 genomic locus in SU-CCS-1 tumor cells. EWSR1-ATF1 (EA) binding sites and H3K27ac ChIP-seq signal are also shown (*middle*). Chromatin loops in the bottom panel are color-coded depending on their association (EA loops, red) or not (Non EA loops, blue) with the fusion protein binding sites.

Supplementary Data 1). Importantly, DNA motifs for MITF, as well as the TFAP and SOX TF families, were also enriched at the "de novo" distal sites shown in Fig. 7a (Fig. 7e). This suggests that upon EWSR1-ATF1 removal CCS cells may undergo differentiation changes driven by a set of TFs also found at the fusion protein binding sites. Consistent with these observations, Gene Ontology analysis of the 157 induced transcripts revealed their enrichment for functions related to NCSC differentiation and chromatin regulation (Fig. 7f and Supplementary Data 1).

Given that CCS typically arise in the deep layer of the skin, to get better insight into the nature of these differentiation changes we interrogated the human skin single cell atlas[32] for primary cell types enriched in either the 417 downregulated direct target genes described in Fig. 6A, or the 157 "de novo" genes from Fig. 7a.

This atlas includes 168,103 single cells derived from five healthy human skin samples, and includes melanocytes and Schwann cells (SchC), which have been previously suggested as putative cell of origin for CCS (Fig. 7g). Interestingly, whereas the EWSR1-ATF1 target gene repertoire was not significantly enriched in any of the 23 different cell types composing this atlas (Fig. 7h, *bottom*), the "de novo" gene signature scored in melanocytes and in stromal SchC (Fig. 7h, *top*). To further investigate the differentiation changes induced by EWSR1-ATF1 depletion, we analyzed siEA- and siCTL-transfected SU-CCS-1 cells with electron microscopy to evaluate the presence of melanosomes. Melanosomes are organelles found in epidermal melanocytes, in which melanin pigments are synthesized and stored, and their presence is associated with terminal melanocytic differentiation.

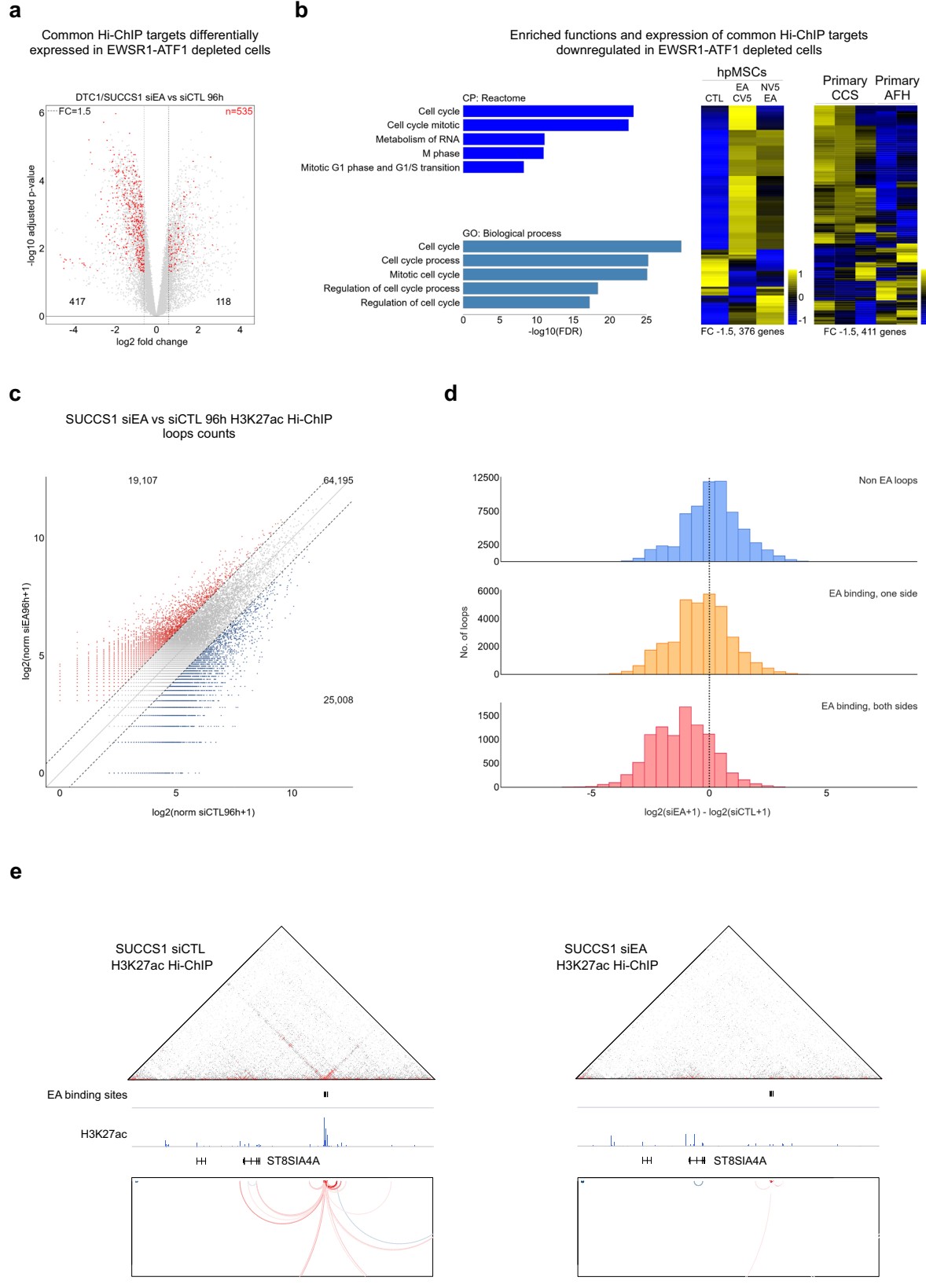

Interestingly, the number of melanosomes per cell was significantly increased in EWSR1-ATF1 depleted cells (p < 0.0001, Supplementary Fig. 6). These results are in agreement with the increased expression of genes involved in melanosome biogenesis and maturation observed in EWSR1-ATF1-depleted cells, supporting the notion that CCS may originate from precursor cells along the neural crest lineage, and suggest that the fusion protein may induce a cellular reprogramming toward a more undifferentiated state by retargeting the lineage-specifying TFs TFAP2A, SOX10, and MITF from their original binding sites.

**Fig. 6 EWSR1-ATF1-connected loops regulate cellular proliferation programs. a** Volcano plot showing gene expression changes in cells with EWSR1-ATF1 depletion (siEA) vs control (siCTL) cells at FC = 1.5 (dashed lines) and adjusted *p*-value = 0.05 (3693 genes). EWSR1-ATF1 direct target genes from Fig. 5e that are differentially expressed at FC = 1.5 (*n* = 535) are marked in red. Two-sided *p*-values are given by lmFit multiple linear fitting models and adjusted for multiple testing by Benjamini & Hochberg correction. **b** *left*: Functional analysis of the 417 genes downregulated upon EWSR1-ATF1 depletion as in (**a**) for Biological Process (GO: BP) and Canonical Pathways (CP: Reactome) showing their involvement in cell cycle regulation. *Right:* Heatmaps depicting z-score expression levels of the downregulated genes from (**a**), in hpMSCs expressing the fusion protein (EA-CV5 and NV5-EA) vs control (CTL), or in primary CCS and AFH tumors. **c** Scatter plot showing the changes in H3K27ac-associated loops between siCTL and siEA transfected SU-CCS-1 cells. The number of loops displaying a more than two-fold change is shown in red (induced) and blue (repressed). **d** Bar plots showing the changes in loop number for EWSR1-ATF1-associated (EA binding, orange and red) or -independent (Non EA, blue) loops. **e** Hi-ChIP and ChIP-seq signal tracks at the ST8SIA4A genomic locus in control (siCTL, *left*) or EWSR1-ATF1-depleted (siEA, *right*) SU-CCS-1 cells.

## Discussion

Oncogenic transcription factors can have variable effects, ranging from senescence and death to transformation, depending on the biological context. The permissiveness of a given cell type to these oncogenes is likely to depend in large part on the pre-existing epigenetic and regulatory landscape. A better understanding of oncogene-driven cellular reprogramming could be instructive toward devising tumor differentiation protocols that synergize with standard chemotherapy and immune-modulatory approaches. Such mechanism-based strategies may be particularly relevant in sarcomas, where onco-fusion proteins are major oncogenic drivers and induce profound changes in chromatin organization and cellular differentiation. Interestingly, CCS bear morphological and molecular similarities with malignant melanomas (MM), suggesting that these two entities may express related TF networks and differentiation programs, some of which may regulate their metastatic proclivity and resistance to conventional therapies.

To define the transcriptional programs controlled by EWSR1-ATF1 in CCS we combined chromatin profiling with nuclear topological mapping to define the regulatory network and direct target repertoire of this fusion protein. We assessed differences in 3D chromatin connectivity and paired these data with single cell expression profiles to study CCS plasticity in the absence of EWSR1-ATF1 and its association with potential cells of origin of these tumors. Remarkably, in contrast to wt ATF1, we find that EWSR1-ATF1 binds mostly distal genomic regions, a feature that was markedly reduced upon mutation of its EWSR1 prion-like domain. Given the ability of EWSR1, FUS and TAF15 to induce phase-separated condensates[33,34], and the critical role of their prion-like domains in establishing DNA accessibility and chromatin activation in other sarcomas[21], our studies suggest that EWSR1-ATF1 may display similar neo-morphic pioneer properties enabling CCS tumor initiation, including the ability to selectively interact with the SWI/SNF remodeling complex.

We further found that EWSR1-ATF1 binding sites are enriched for additional TF motifs, including the well-known neural crest regulators TFAP2A and SOX10, and are co-occupied by MITF, a master regulator of melanocyte development and cellular pigmentation. Cooperation with these factors may serve a dual function, recruiting them to EWSR1-ATF1 sites for transcriptional activation, and displacing from their original binding sites to further alter the differentiation state of the tumor precursor cells. Remarkably this also applies to wt ATF1. Physiological expression of ATF1, and the related CREB and CREM proteins, is required during murine and human embryonic development[35–37], whereas gains in ATF1 expression and/or activity are associated with increased tumor cell survival, proliferation, and dissemination in different cancer types (Chen M et al, in press, https://doi.org/10.1016/j.gendis.2021.04.008). Consistent with this notion, the recruitment of wt ATF1 by the fusion protein to cell-type restricted distal sites may alter its activity and favor CCS development. Although we cannot rule out

the possibility that wt ATF1 may, in some instance, be indirectly recruited to EWSR1-ATF1 binding sites as a consequence of the increases in chromatin accessibility exposing CRE-motifs, the reduction in ATF1 N signal observed upon EWSR1-ATF1 depletion speaks in favor of a direct dependency on EWSR1-ATF1 binding. Other CRE-motif binding TFs may also collaborate with EWSR1-ATF1 and modulate its activity, given the heterodimerization capabilities of this bZIP family of TFs.

TFAP2, SOX10 and MITF are involved in the early steps of NC induction, lineage specification, and terminal differentiation[27], and alterations in their expression are associated with the metastatic behavior of NC -derived tumors[38,39] as well as other cancers. In MM for example, gain in ATF1 and SOX10, associated with loss of TFAP2A expression, drive tumor growth and metastasis by regulating, in an opposing manner, a number of genes involved in cell adhesion, extracellular matrix remodeling, and cell survival[38,39]. Given the constitutively active nature of EWSR1-ATF1, it is tempting to speculate that the expression of the fusion protein in CCS, combined with the genomic retargeting of TFAP2A, SOX10, and MITF, may sustain an invasive phenotype that mimics MM progression. This notion is supported by the observation that CCS have the highest frequency of lymph node metastases among sarcomas[40], most of which tend to disseminate by the hematogenous route. TFAP2A and MITF also functions as gatekeepers that control the balance between cellular proliferation and differentiation, two typically mutually exclusive processes[41–43]. One plausible scenario is that in CCS the cooperative binding of EWSR1-ATF1 with additional TFs may tilt the balance toward one of the two processes, with EWSR1-ATF1 depletion resulting in the opposite effect. Thus, the collaborative TF network identified in this study may serve EWSR1-ATF1 to induce the oncogenic reprogramming required for tumor development.

A relevant finding in our study is the significant changes in 3D interactions that follow EWSR1-ATF1 depletion, despite the relatively limited number of binding sites of the fusion protein. Importantly, these changes include both chromatin looping gains and losses, suggesting that these alterations in cis-connectivity may result from the combined decrease in EWSR1-ATF1 direct transcriptional targets and the increase in differentiation genes. Gaining the ability to profoundly impact chromatin interactions may even represent a potential key feature of sarcoma-promoting fusion proteins like EWSR1-FLI1 and EWSR1-ATF1, which provides the establishment of permissive oncogenic programs and differentiation states.

Our study highlights the power of combining changes in chromatin activity and connectivity with primary single-cell expression profiles to uncover differentiation states that define tumor and progenitor cell identities. In CCS, genetically-engineered mouse models suggested MSCs and NCSCs to be particularly permissive toward EWSR1-ATF1 driven transformation[9,10]. More recently, the CCS cell-of-origin was proposed to be a NC-derived peripheral nerve cell expressing the

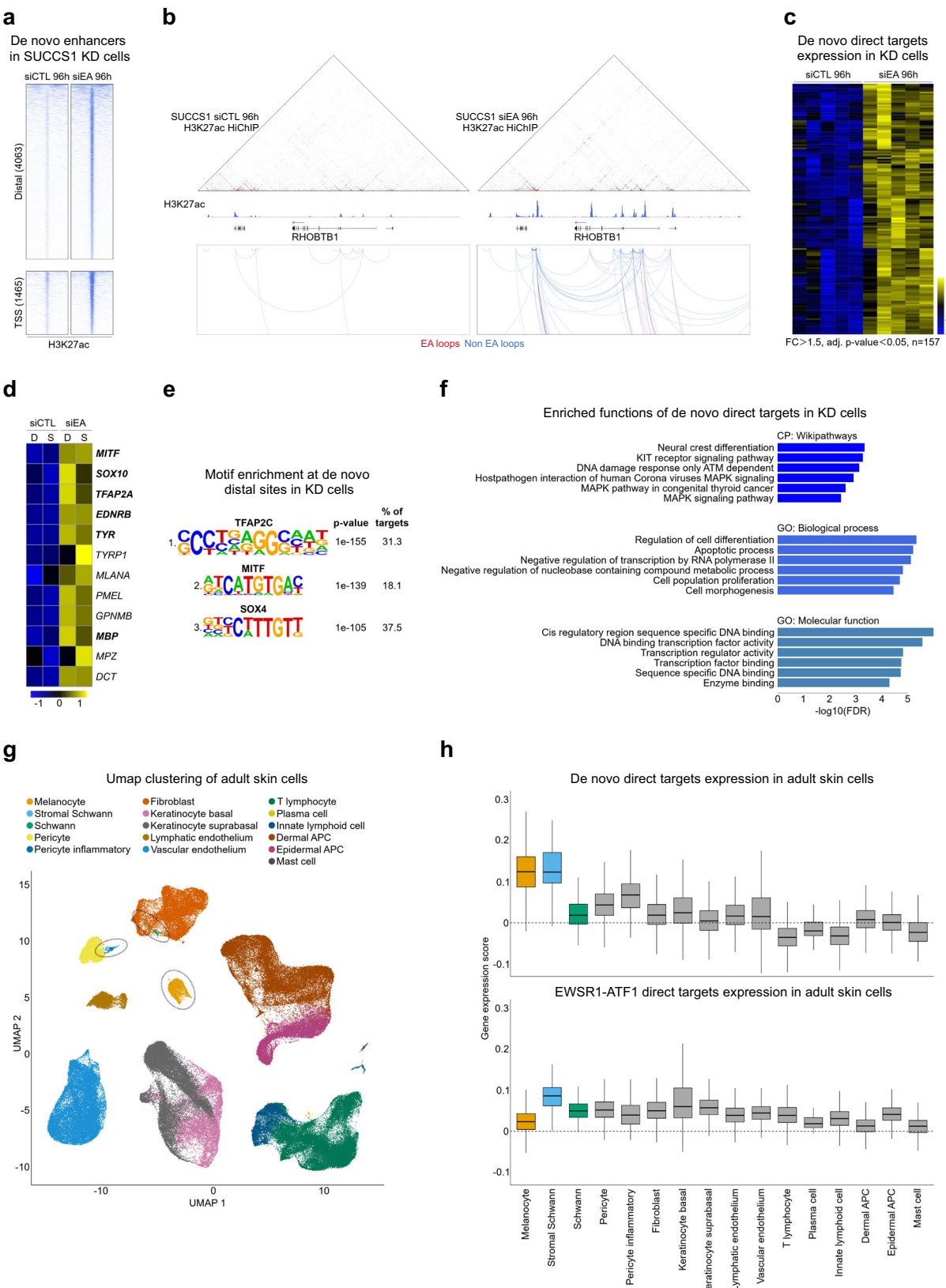

*Mpz* gene, involved in myelin production and normally restricted to SchC[8]. Although these results are consistent with the expression of SchC markers in human CCS, in the same study EWSR1-ATF1 expression in terminally differentiated SchC and melanocytes failed to produce tumors, suggesting that CCS may arise from less differentiated neural crest derivatives, and that

the SchC-related phenotype may be conferred by the fusion protein.

Consistent with these observations, our functional analysis of the transcriptional signature induced de novo in EWSR1-ATF1-depleted CCS cells showed strong enrichment in terms related to neural crest differentiation and chromatin regulation.

**Fig. 7 Changes in 3D connectivity upon EWSR1-ATF1 depletion point to candidate CCS precursor cells. a** Heatmaps depicting H3K27ac signal at Distal (*top panels*) and Proximal (*bottom panels*) sites in SU-CCS-1 cells, transfected with either control (siCTL, *left*) or EWSR1-ATF1 targeting (siEA, *right*) siRNAs, illustrating the induction of "de novo" sites in EWSR1-ATF1-depleted cells. For each heatmap 20 kb regions centered on the H3K27ac signal are shown. Signals are ranked by H3K27ac intensity in siEA-transfected SU-CCS-1 cells. **b** Hi-ChIP interaction map (*top*) and chromatin looping profile (*bottom*) for the RHOBTB1 genomic locus in siCTL or siEA – treated SU-CCS-1 tumor cells. H3K27ac ChIP-seq signal are also shown (*bottom*). Chromatin loops in the bottom panel are color-coded depending on their association (EA loops, red) or not (Non-EA loops, blue) with the fusion protein binding sites. **c** Gene expression z-scores heatmaps in DTC1 and SU-CCS-1 cells for the 157 transcripts identified as targets of "de novo" regulatory sites as in (**a**), showing upregulation at FC > 1.5 and p-value<0.05 in both cell lines upon EWSR1-ATF1 depletion. **d** Expression z-scores of genes involved in melanocytic differentiation, showing a higher expression in siEA-transfected DTC1 (D) and SU-CCS-1 (S) cells. Genes marked in bold were among the de novo targets in (**c**). **e** Homer motif analysis of the 4063 "de novo" distal sites shown in (**a**). Binomial p-values are given by the motif enrichment software HOMER. **f** Functional analysis of the same 157 genes for the Wiki-Pathways (WP), Biological Processes (GO: BP), and Molecular Function (GO: MF), showing the involvement of this gene repertoire in the NCSC differentiation and chromatin remodeling. **g** UMAP representation for the expression profiles of 168,103 single cells derived from five healthy human skin samples (including both epidermis and dermis layers). Normalized log2 scores were used for UMAP generation. Melanocyte, Schwann and Stromal Schwann cells are marked with dashed circles. **h** Boxplots depicting the enrichment score for the downregulated EWSR1-ATF1 target genes (*bottom*, n = 393) or the upregulated "de novo" gene signature (*top*, n = 154) expressed across 16 different cell groups. Following the removal of the fusion protein the tumor cells acquire a more differentiated phenotype, reminiscent of melanocytes and stromal Schwann cells. Box plots show the 1st to 3rd quartiles (25–75%) of the data with median values indicated.

Interestingly, among the induced genes we found *TFAP2A*, *SOX10*, *MITF*, *TYR* and *MBP*, all of which are involved in terminal melanocytic differentiation of NC-derivatives, with SOX10 and TFAP2A being required to induce MITF expression[43]. *MBP* is a myelination gene upregulated during SchC terminal differentiation and repressed during the SchC reprogramming associated with nervous tissue regeneration[44]. Although not a direct target of de novo enhancers, *MPZ* was also upregulated upon EWSR1-ATF1 depletion in CCS tumor cells. Because this gene was previously identified as a marker of CCS precursor cells, it is conceivable that the cell-of-origin of these tumors lies on a differentiation path between SchC precursors and melanocytes, whose trajectory becomes modified by the topological and transcriptional changes induced by EWSR1-ATF1. Since SchC and a subset of bone-marrow derived MSC have been proposed to share a common NCSC origin[45], it is tempting to speculate that their similar developmental ontogenesis underlies their permissiveness toward EWSR1-ATF1-driven transformation, and the emergence of EWSR1-ATF1-associated tumors in multiple different anatomical locations, including the bone. In agreement with this hypothesis, our 157 de novo gene signature was found to be enriched in single-cell profiles of differentiated melanocytes and SchC in a skin dataset from the Human Cell Atlas (HCA) that included more than 160,000 primary single cell analyses[32]. In contrast, EWSR1-ATF1 direct target genes were only moderately enriched in SchC profiles. Remarkably, the loss of the fusion protein was followed by a marked increase in the number of melanosomes in SU-CCS-1 cells, confirming the induction of terminal melanocytic differentiation in these cells. Taken together, these results suggest that CCS precursor cells retain the plasticity to differentiate along the melanocytic and SchC lineages upon EWSR1-ATF1 depletion.

Together with their potential common origin from NC derivatives, our study provides further molecular insight into the biological similarities between MM and CCS. MM is among the deadliest forms of cancer, mainly due to its high metastatic potential. The acquisition of its invasive and metastatic phenotype has been associated with cellular de-differentiation and the hijacking of developmental programs that characterize NCSCs[46]. Recently, the mechanisms of resistance toward combined BRAF and MEK inhibitors in MM have also been associated with the emergence of a transient population of tumor cells that display NSCS properties, confirming the importance of non-genetic drivers of drug tolerance in these tumors[47]. Similarly, the de-differentiated state induced by EWSR1-ATF1 may promote intrinsic resistance to conventional therapies in CCS, which

remains the major obstacle for their effective treatment. Given that this reprogramming process may be driven by the genomic retargeting of SOX10, TFAP2A and MITF to EWSR1-ATF1 binding sites, retargeting these TFs to induce a melanocytic phenotype could constitute an effective therapeutic strategy, allowing tumor cells to differentiate and become more chemosensitive and possibly immunogenic.

Interestingly, our data on the chromatin and expression profiles of AFH suggest that these tumors arise from different precursor cells despite sharing the same oncogenic driving event. In support to this notion, expression of EWSR1-CREB1 and EWSR1-ATF1 in undifferentiated human ESCs leads to gene expression signatures that recapitulate AFH and GI-CCS, respectively, whereas expression of EWSR1-ATF1 in ESCs-derived MSCs induces the CCS melanocytic markers SOX10, PMEL and SLC7A5[48].

Collectively, our observations draw connections between cellular histogenesis and molecular phenotypes in EWSR1-ATF1-associated tumors. They also suggest that the differentiation of these tumors could be modulated to achieve a cellular state that is more responsive to current therapies. However, different phenotypes may co-exist among cell subpopulations in the same tumor sample and their aggressiveness may be determined, at least in part, by different expression levels of the fusion protein and its collaborative TFs, as recently reported for EWSR1-FLI1 in Ewing sarcoma[49]. This hypothesis could be addressed by single-cell expression profiling of primary tumors. In summary, although rare, CCS may provide a valuable model to decipher the molecular underpinnings that link cellular differentiation to tumor development and therapeutic resistance in other cancer types. In particular, the noteworthy similarity between CCS and MM delineated here may pave the way for the identification of new differentiation therapies that render both tumors, as well as other EWSR1-ATF1-associated diseases, more sensitive to available therapies.

## Methods

**Cell culture and primary tumors**. CCS and AFH tumor specimens and hpMSCs were collected with approval from the Institutional review Boards (IRB) of Centre Hospitalier Universitaire Vaudois (CHUV, University of Lausanne, Switzerland), Massachusetts General Hospital (MGH, Boston, USA) and Skåne University Hospital (Lund University, Sweden). Samples were anonymized prior to analysis. Informed written consent was obtained for all prospective samples collected for this study. Tumor samples collected before 2015 were obtained from the tumor bank under the same IRB for discarded material.

hpMSCs obtained from healthy patients were cultured in IMDM (Gibco) supplemented with 10% FBS (Pan biotech), 1% MEM NEAA (Gibco), 1% Pen/Strep (Gibco) and 10 ng/ml PDGF-BB (Prospec) as described previously[50].

The CCS cell lines SU-CCS-1 (ATCC CRL-2971)[51], and DTC1[52] kindly provided by Professor T. Nielsen (Department of Pathology and Laboratory Medicine, University of British Columbia, Vancouver, Canada), were cultured in RPMI-1640 (Gibco) supplemented with 10% FBS and 1% Pen/Strep as suspension in Ultra-low attachment flasks (Corning). Cells were split 2–3 times/week, whereby spheres were dissociated by pipetting, to keep a density of around 10 000–50 000 cells/cm². HEK293T cells (ATCC CRL-1573) were cultured in DMEM + GlutaMAX (Gibco) supplemented with 10% FBS, 1% MEM NEAA and 1% Pen/Strep. All cell cultures were maintained at 37 °C and 5% $CO_2$ in a humified chamber.

**Construction of expression plasmids.** The full-length coding sequence (CDS) of type 1 EWSR1-ATF1 transcript was amplified from DTC1 cDNA with the Phusion High-Fidelity DNA polymerase (Thermo Fisher) and the primers EWSR1_269F and ATF1_1313R, and subsequent nested PCR with the primers EWSR1_296F and ATF1_1245R, according to the manufacturer's protocol. The full-length CDS of the wt ATF1 transcript was amplified from HeLa cDNA using the primers ATF1_216F and ATF1_1493R. All primer sequences are listed in Supplementary Table 1. The amplified fragments were purified from agarose gel and subcloned with the TOPO TA cloning kit according to the manufacturer's instructions (Invitrogen). The V5 tag sequence was added in-frame to the 5′ or 3′ end of the EWSR1-ATF1 CDS by Phusion PCR using the primers EWSR1_V5_LR_F and ATF1_LR_R (N-terminal V5 tag, NV5-EA) or EWSR1_LR_F and ATF1_V5_LR_R (C-terminal V5 tag, EA-CV5). The V5 tag was added in-frame to the 3′ end (ATF1-CV5) of the wt ATF1 CDS using the primers ATF1_LR_F and ATF1_V5_LR_R. The V5 tag encodes the predicted amino acid (aa) sequence GKPIPNPLLGLDST. The amplified fragments were purified from agarose gel, subcloned using the pENTR/SD/D-TOPO cloning kit (Invitrogen), and then subsequently cloned into the lentiviral vector pLIV[53] using the Gateway LR Clonase II according to the manufacturer's instructions (Invitrogen). All steps of the cloning process were sequence verified.

For the generation of the EWSR1(YS37)-ATF1 mutant construct (E(YS37)A-CV5) a C-terminally V5-tagged EWSR1-ATF1 clone was first synthesized as gBlocks™ Gene Fragments (IDT, Integrated DNA technology), with codon optimization for the human tRNA as proposed by the manufacturer, amplified and subsequently cloned with the pENTR/D-TOPO cloning kit. The E(YS37)A-CV5 mutant which has all the 37 tyrosine residues of the EWSR1 prion-like domain replaced by serine[21], was amplified with the Phusion High-Fidelity DNA polymerase and the primers EWSR1_mut_ATF1_F and EWSR1_mut_ATF1_R (Supplementary Table 1). The mutant sequences were cloned into pENTR/D-TOPO and subsequently into pLIV as described above.

**Knock-down of EWSR1-ATF1 by siRNA transfection.** EWSR1-ATF1 type 1 breakpoint specific siRNAs (siEA, sense GCGGUGGAAUGGGAAAAAUUUUU; antisense 5′ P-AAAUUUUUCCCAUUCCACCGCUU, sequence adapted from Yamada et al., 2013) (ON-TARGETplus siRNA, Dharmacon), or non-targeting siRNAs (Allstars negative control siRNA, Qiagen), were transfected using the Lipofectamine RNAiMAX transfection reagent (Thermo Fisher). For this, siRNA and reagent were mixed in RPMI medium without supplements at the ratio 900:0.025 (pmol siRNA:μl reagent) per T-75 flask and incubated 20 min. Transfection complexes were added to cells seeded at 24,000 (DTC1)–26 660 (SU-CCS-1) cells/cm² and the cell medium was changed the day after transfection. The cells were harvested at 72 h and 96 h after transfection for RNA and protein extraction, and for chromatin-protein cross-linking. The KD efficiency was evaluated by RQ-PCR and western blot analyses.

**Lentiviral infection for wt ATF1 depletion and EWSR1-ATF1 induction.** For wt ATF1 depletion, a pLKO.1 lentiviral shRNA targeting the 5′ part of the wt ATF1 transcript (Clone ID: TRCN0000273888) was purchased from the RNAi Consortium. SU-CCS-1 cells were infected with shATF1-expressing lentiviral construct, or control GFP targeting (GCAAGCTGACCCTGAAGTTCAT) construct. For EWSR1-ATF1 over expression, low passage (p4-7) hpMSCs were infected with the EA-CV5, NV5-EA, mutant E(YS37)A-CV5, wt ATF1-CV5 or empty pLIV lentiviral expression plasmids. Briefly, each construct was co-transfected with envelope and packaging plasmids (pMD2.G and pCMV-ΔR8, Addgene) into 293 T packaging cells using FuGENE 6 Transfection reagent (Promega) for virus production. After 72 h the viral supernatant was collected and concentrated using the Lenti-X concentrator reagent (Clontech) according to the manufacturer's instructions. The cell medium was changed the day after infection. Infected SU-CCS-1 cells were harvested after 3 days, infected hpMSC cells were selected by adding 0.75 μg/ml Puromycin (InvivoGen) after 4 days and then harvested after 8 days. The level of expression for each gene and corresponding protein was evaluated by RQ-PCR and western blot analyses, respectively.

**RQ-PCR, Western blot, co-immunoprecipitation, b-isox precipitation and electron microscopy analyses.** RNA was extracted with the RNeasy Mini kit (Qiagen) and typically 500 ng of total RNA used for each cDNA reaction. RQ-PCR was performed as described[50] using the PowerUp SYBR Green mix (Applied Biosystems) and specific primers (Supplementary Table 1). Three replicates of each condition were used to calculate mean Ct values, mean expression values were then calculated according to the comparative $2^{-\Delta\Delta Ct}$ method (Applied Biosystems) relative to the control sample and normalized to the endogenous control GAPDH.

The antibodies used for western blot analysis were anti-ATF1 C (1:500, LS-C351857, LSBio or ab134104, Abcam), anti-BRG1 (1:1000, E906E, Cell signaling) anti-V5 epitope (1:5000, 46–1157, Invitrogen), anti-α-Tubulin (1:1000, CP06, Millipore) and HRP-conjugated sheep anti-mouse (1:5000, Cytiva) or goat anti-rabbit secondary antibodies (1:20 000, Dako). Whole-cell protein lysates were prepared by standard methods and quantified using the Bradford reagent (Bio-Rad). Protein electrophoresis was performed in Tris-Glycine 12% SDS-Polyacrylamide gels or pre-cast Bolt 4–12% Bis-Tris gels in MES running buffer and Mini gel tank according to the manufacturer's protocol (Invitrogen). Blotting was performed by standard procedure, membranes developed using the ECL substrates SuperSignal West Pico PLUS (Thermo Fisher Scientific) or WesternBright Sirius (Advansta) and proteins visualized with a Fusion FX camera (Vilbert-Loumat).

Co-IP experiments were performed as previously described[21,54] using 500 μg of protein from whole cell lysates and the 2 μg antibody (unless otherwise specified); anti-ATF1 C (LS-C351857, LSBio or ab134104, Abcam), -ATF1 N (A303-034A, Bethyl laboratories), -BRG1 (6 μl) or −V5 epitope antibodies. Western blot analysis was performed as described above using the anti-ATF1 C, -BRG1, −V5, or anti-α-Tubulin antibodies. Beta-Isox (biotin-isoxazole) aggregation experiments were performed as previously described[21].

1M siEA- and siCTL-treated SU-CCS-1 cells were fixed in a mixture of 2.5% paraformaldehyde and 2% glutaraldehyde in 0.005 M sodium cacodylate buffer (pH 7.4), and then treated with 1% osmium tetraoxide in 0.1 M sodium cacodylate buffer (pH 7.4). After dehydration in graded series of ethanol, pellets were transferred to propylene oxide and embedded in Epon-Araldyte. 100 nm thick sections were cut and mounted on nickel grids covered with Formvar membranes, stained with uranyl acetate and lead citrate, and examined with a Morgagni 268D Transmission electron microscope (Philips, Eindhoven, The Netherlands). Melanosomes were counted in 30 cells for each condition, and statistical significance evaluated by t-test.

**Transcription factor (TF) and histone mark ChIP-seq analysis.** DTC1, SU-CCS-1 and hpMSCs, $2–10 \times 10^6$ cells per sample and epitope, were used for ChIP following previously described procedures[55,56]. Primary CCS and AFH tumors were processed as described[57]. In short, cells or tissue fragments were cross-linked with formaldehyde, chromatin was fragmented using a Branson 250 sonifier and then immunoprecipitated overnight with 1 ug antibody/million cells of the anti-ATF1 C (LS-C351857, LS Bio), -ATF1 N (A303-034A, Bethyl laboratories), -EWSR1 N (sc-48404, Santa Cruz), -TFAP2A (sc-12726, Santa Cruz), -SOX10 (PA5-40697, Invitrogen), -MITF (#91201, Active Motif), -H3K4me3 (#07–473, Millipore), -H3K27ac (#39133, Active motif), -H3K4me1 (ab8895, Abcam), -p300 (#54062, Cell signaling) or -V5 (#13202, Cell signaling) antibodies. Antibody-chromatin complexes were pulled down using Protein G Dynabeads (Life technologies), washed and eluted. Immunoprecipitated DNA was treated with RNAse A and Proteinase K, purified using Agencourt AMPure XP beads (Beckman coulter) and quantified with the Qubit dsDNA HS assay (Thermo Fisher). Sequencing libraries were prepared from ChIP DNA and Input DNA controls with the TruSeq ChIP Sample preparation kit and the SetA or SetB adapters (Illumina) according to the manufacturer's protocol, with the addition of performing a 200–500 bp size selection of the libraries utilizing AMPure XP beads. Sequencing was performed on a Hi-Seq Illumina Genome Analyzer (50 bp single-read).

**TF and histone mark ChIP-seq data analysis.** Single-end reads were trimmed using TrimGalore version (v) 0.6.4 (https://github.com/FelixKrueger/TrimGalore) and aligned to the human genome assembly GRCh37 (hg19) using STAR v 2.5.0a[58]. Multiple matches, as well as regions present in the ENCODE project DAC Exclusion List Regions (data set ENCSR636HFF), were removed and maximum three redundant reads kept for subsequent analyses.

Significantly enriched genomic regions for TF and histone mark ChIP-seq were detected using MACS2 v 2.2.6[59]. Multiple FDRs were screened until reaching the NR requirements, giving the FDR of 1e-7 for ATF1 C, H3K4me3, TFAP2A, SOX10, MITF, ATF1 N, and V5, and 1e-4 for EWSR1 N. To reduce the amount of false-positive enrichment, we applied another method to identify significantly enriched genomic regions[60]. Briefly, the whole genome was split into 400-bp bins and the scores (log2 of the bins reads counts) of these regions were calculated for both the IP and the Input samples. A pseudo count was added to avoid NAs. Bins were then split into quantiles based on the mean scores of the IP and Input ((IP + Input)/2) and for each quantile Z-scores were calculated using the mean and SD of the bins fold change (log2(IP)-log2(Input)) within the quantile. P-values were calculated using the pnorm function in R and adjusted with the Benjamini-Hochberg method. In each quantile, bins with an FDR below 0.001 were considered as enriched. Only the MACS2 peaks that intersected with the enriched genomic regions identified by the above-mentioned method were kept for further analysis. The BEDtools v 2.27.1 intersect command, with the u-option of unicity, was used to extract the enriched regions common between the two methods[61]. This process was applied to each ChIP-seq sample separately.

For the generation of the EWSR1-ATF1 reference peak set, all ATF1 C peaks identified in the four ChIP-seq experiments (two replicates each of the DTC1 and

SU-CCS-1 cell lines) were merged, as well as regions overlapping within +/− 150 bp. Regions enriched in all four experiments were kept for further analyses. The same procedure was applied to the four EWSR1 N ChIP-seq experiments. 7402 ATF1 C peaks were identified of which 2385 overlapped with EWSR1 N peaks, constituting the EWSR1-ATF1 reference peak set used for subsequent analysis. 1288 ATF1 C peaks that displayed a complete lack of EWSR1 N correspond to sites bound by wt ATF1 only. The peaks within +/− 1 kb of a known gene's TSS (ucsc_RefFlat_07_08_2016) were labeled as TSS. The remnant peaks were evaluated for H3K4me3 enrichment as above and labeled as TSS if H3K4me3 enriched regions were identified in four ChIP-seq experiments, giving in total 374 TSS-associated EWSR1-ATF1 peaks. All the other peaks were labeled as Distal ($n = 2011$). For the genomic distributions, the Distal peaks were annotated as intragenic or intergenic according to the ucsc_RefFlat_07_08_2016 list. IGV was used to visualize ChIP-seq tracks[62].

CPM-normalized BigWig files were generated using bamCoverage[63], with an extension of ChIP-seq reads to 300 bp fragment length, and used for the ChIP signal heat maps and composite plots (generated with the deepTools scripts computeMatrix, plotHeatmap and plotProfile). Heat maps were sorted based on the ATF1 C mean signal in the four cell line replicate ChIP-seq experiments.

For ChIP-seq peak score calculation, the read counts per peak was calculated using bedTools with the −c option, normalizing to 20 M reads per sample (the approximate mean reads per sample) and scaling to 500 bp peak width (the approximate EWSR1-ATF1 reference peak width). Each peak score is given by $\log2(IP + 1)-\log2(Input + 1)$ where +1 is an additional pseudo-count, mean values of the replicates. For each data set, the mean peak read count in the input sample was used as the lower threshold. The peaks in the IP sample with a read count below the threshold were set to zero. Correlations between ChIP-seq scores were calculated by Pearson's correlation coefficient ($r$).

Centered, 500 bp extended peak regions were searched for De Novo motif enrichment using HOMER v 4.11.1[64], applying the 500 bp target sequence length and chopify options. The De Novo motifs were then clustered using the «compareMotifs.pl» script from HOMER. To find motif half-site enrichment, a second round of motif analysis was performed using shorter motif length (4, 5, 6 nt).

Analysis of the ENCODE data sets of wt ATF1 ChIP-seq profiling in HepG2 [https://www.encodeproject.org/experiments/ENCSR253OON/] and K562 [https://www.encodeproject.org/experiments/ENCSR091GVJ/] cells (ENCODE Project Consortium, 2012)[65] was performed as described above. The peaks in replicates of the same cell line were pooled and annotated as TSS or Distal as described. External raw data files were acquired using fastq-dump of the SRA toolkit v 2.10.3 (https://github.com/ncbi/sra-tools).

For defining a set of control TSS sites, H3K27ac peak scores for the siEA and siCTRL IPs of each cell line was calculated as described above on Hg19 non-overlapping protein-coding genes +/− 500 bp around the TSS. The mean score between siEA and siCTRL was calculated and the top 374 top-scoring genes was extracted and used for generating composite plots of H3K27ac and ATAC signals.

To define H3K27ac changes in EWSR1-ATF1 depleted cells, H3K27ac peaks in DTC1 and SU-CCS-1 siEA and siCTL-treated cells were first combined (with a siEA or siCTL sample tag), then peaks within 150 bp were pooled. Distal/TSS annotation and H3K27ac ChIP peak scores calculation was performed as described. This analysis yielded 617 and 5872 peaks at $\log2FC > 2$ cutoff, and 2209 and 4942 eaks at $\log2FC<-2$ cutoff, in DTC1 and SU-CCS-1 cells, respectively. The "de novo" H3K27ac peaks in SU-CCS-1 siEA were defined as having <5 reads counts in siCTL and a FC > 4 in siEA. This approach generated a total of 5528 peaks, of which 4063 were Distal and 1465 TSS-associated.

The hpMSC ChIP-seq data were analyzed as described above. The EA-CV5 and NV5-EA V5 peaks that overlapped with the 2385 reference peak set, and that were not present in the CTL samples, were selected for further analyses, giving 1432 peaks in EA-CV5 and 1322 peaks in NV5-EA. The V5 peaks were subcategorized based on ATF1 N peak presence; the No ATF1 peak category lacking ATF1 N peaks in both CTL and EA-CV5 ($n = 449$)/NV5-EA ($n = 495$) samples, the De Novo peak category having ATF1 N peaks in only EA-CV5 ($n = 398$)/NV5-EA ($n = 237$) samples and the Pre-existing peak category having ATF1 N peaks in both CTL and EA-CV5 ($n = 585$)/NV5-EA ($n = 590$) samples. The pre-existing peak category was further evaluated by ATF1 N reads counts scores calculation, and only the sites showing an ATF1 N score increase in EA-CV5 ($n = 475$)/NV5-EA ($n = 427$) were kept for further analysis. Out of the No ATF1, De Novo, and Pre-existing sites, 395 (88%), 340 (85%), and 364 (77%), respectively, were distal and 54, 58, and 111, respectively, were TSS-associated. For the EA-CV5, E(YS37)A-CV5, and ATF1-CV5 hpMSC ChIP-seq experiment, all V5 peaks of the three samples were first combined with a sample tag (EA-CV5, E(YS37)A-CV5 or ATF1-CV5) added to each peak and peaks within 150 bp were pooled. To obtain the same number of V5 peaks in each sample and compare the genomic distributions of only the strongest peaks, reads counts scores were calculated for each peak ($\log2(IP + 1)-\log2(Input + 1)$) and quantile normalized across samples, and only peaks above score 3.5 were considered. Annotation of peaks as TSS or Distal was performed as described above. This analysis obtained 16487 V5 peaks in all three samples, whereof 15308 (93%), 7293 (44%) and 6545 (40%) were distal in EA-CV5, E(YS37)A-CV5, and ATF1-CV5, respectively, and 1179 (7%), 9194 (56%) and 9942 (60%) were TSS-associated.

All figures, except heat maps and composite plots, were generated using R (R: A language and environment for statistical computing. R Foundation for Statistical Computing, Vienna, Austria, http://www.R-project.org/).

**ATAC-seq and DNAse analysis**. DTC1, SU-CCS-1, and EA-CV5/NV5-EA/empty pLIV infected hpMSCs, 75,000 cells/reaction, were used for ATAC-seq analysis following a procedure adapted from a published protocol[66]. Briefly, the cells were washed once in PBS and then re-suspended in 1 ml Lysis buffer without Igepal to swell the cytoplasm, and centrifuged 10 min at $500 \times g$, 4 °C. The resultant nuclei preparation was re-suspended in 250 μl Lysis buffer with Igepal, incubated 10 min on ice, and then centrifuged 10 min at $500 \times g$, 4 °C. Tagmentation, DNA purification and PCR amplification (12 cycles) were performed according to the previous protocol, and sequencing was performed on a Hi-Seq Illumina Genome Analyzer (50 bp paired-end). Paired-end reads were aligned as described above and positions readjusted to compensate for the transposase 9 bp shift; reads aligning to the + strand were offset by +4 bp and reads aligning to the – strand were offset by −5bp.

The DNAse-seq profiles were downloaded from GEO [https://www.ncbi.nlm.nih.gov/geo/query/acc.cgi?acc=GSE29692], and the DNase signal calculated by averaging signal around 1 kb of peaks (−500 to 500 bp), using a single replicate from each cell type.

**H3K27ac Hi-ChIP analysis**. HiChIP was performed according to a published procedure[67] with a few modifications. Briefly, $5 \times 10^6$ formaldehyde cross-linked cells were lysed in HiC lysis buffer (10 mM Tris-HCl pH8.0, 10 mM NaCl, 0.2% Igepal) for 30 min at 4 °C. Nuclei were washed with HiC lysis buffer, resuspended in 0.5% SDS (100 μl), and incubated for 10 min at 62 °C. The SDS was quenched with 50 μl of 10% Triton X-100 and 265 μl water for 15 min at 37 °C. The chromatin was then digested using 200U of the MboI restriction enzyme in NEBuffer 2.1 (New England Biolabs) for 2 h at 37 °C, followed by heat-inactivation at 65 °C for 20 min. The digested chromatin was end-filled in 1X NEBuffer with 37.5 μl of 0.4 mM biotin-14-dATP (Thermo Fisher), 50U of Klenow polymerase I (New England Biolabs), and a combination of dCTP, dGTP and dTTP (10 mM, 1.5 μl each) for 1.5 h at 37 °C. End-filled chromatin was subjected to proximity ligation (1X ligation buffer, 4000U T4 DNA Ligase, 125 μl of 10% Triton-X100, and 7.5 μl of BSA in 1.5 ml reaction) for 4 h at 25 °C, collected by centrifugation and proceeded with ChIP. Chromatin was lysed in Nuclei Lysis buffer (10 mM Tris, 1% SDS, 10 mm EDTA, 1x protease inhibitor cocktail-PIC), diluted in ChIP dilution buffer (20 mM Tris 7.4, 0.1%SDS, 0.12% DOC, 160 mM NaCl, 1.2% Triton, 1x PIC) and sonicated in a total volume of 800 μl. Immunoprecipitation was performed over night with 5 μg anti-H3K27ac antibody as described above. Hi-ChIP DNA was purified using the ChIP DNA Clean & Concentrator kit (Zymo research) and biotin pulldown performed with 2 μl of 10 mg/ml Dynabeads MyOne StreptavidinT1 beads (Thermo Fisher) in 1X binding buffer (5 mM Tris-HCl, 0.5 M EDTA, 1 M NaCl) for 15 min (50 μl reaction), followed by washes in 1X Tween Washing Buffer (5 mM Tris-HCl, 0.5 mM EDTA, 1 M NaCl, 0.05% Tween 20) and magnetic separation. Sequencing libraries were prepared from Hi-ChIP DNA using the Next Gen DNA Library kit (Active Motif) according to the manufacturers' instructions, carrying out the reactions on bead-bound DNA followed by magnetic separation. The Hi-ChIP DNA library was PCR amplified (8 cycles), purified using AMPure XP beads, and sequenced on a Hi-Seq Illumina Genome Analyzer (100 bp paired-end to 300 M read-pairs/sample).

For the analysis of H3K27ac Hi-ChIP data, paired-end reads were aligned to the hg19 genome using the HiC-Pro pipeline (v 2.7.6). Default settings were used to align paired reads, identify valid interactions and generate interaction matrices. The hichipper tool (v 0.7.7) was used for loop calling by H3K27ac ChIP-seq peaks ($q$-value < = 0.00001), applying the following parameters: -max-distance 100,000,000 -read-length 100. Loops with counts greater than 5 were kept for further analysis. Genes with a promoter region (±1 kb of the TSS) overlapping with loop anchors were identified as EWSR1-ATF1 targeted genes.

To identify chromatin interaction changes following EWSR1-ATF1 depletion in SU-CCS-1 cells, we started generating the union of H3K27ac ChIP-seq peaks from siCTL and siEA SU-CCS-1 cells, and used this peak set to call loops using the hichipper tool. Loops were then normalized using the DESeq2 method and analyzed using the DiffLoop tool. Increased/decreased loops defined using a two-fold change between the siEA and siCTL SU-CCS-1 Hi-ChIP profiles.

**RNA-seq differential expression analysis**. For RNA-seq analysis, total RNA was extracted from frozen tumor tissue using the TRIzol reagent (Invitrogen) or from cells using the RNeasy Mini kit (Qiagen) according to the manufacturer's instructions. Minimum 100 ng used for preparing sequencing libraries using the TruSeq mRNA stranded kit (Illumina) and sequencing was performed on a Hi-Seq Illumina Genome Analyzer (100 bp single-read). Samples used for RNA-seq profiling included siEA- or siCTL-treated DTC1 and SU-CCS-1 cell lines, EA-CV5-, NV5-EA- or empty pLIV infected hpMSCs and the primary tumors CCS1, CCS2, CCS3, AFH1 and AFH2.

Single-end reads were trimmed using TrimGalore v 0.6.4 and aligned to the human genome and transcriptome assembly GRCh37 (hg19) using STAR v 2.5.0a. Transcript quantification was done using rsem v 1.3.0[68]. The resulting counts matrix was used for subsequent analysis. Analysis was done in R; gene filtering based on the rule of 1 count per M (cpm) in at least 1 sample, library size scaling using TMM normalization (EdgeR package v 3.32.1)[69], Robinson Bioinformatics (2010)) and log-transformation using the Limma voom function[70]. Normalized data were batch corrected using the Limma removeBatchEffect function and

unwanted variation was removed with RUVr (RUVseq package v 1.24.0)[71]. Statistically significant differentially expressed genes were identified from log-transformed, TMM-scaled values using lmFit of the Limma package by fitting a linear model on the selected samples. P-values were adjusted for multiple testing using the Benjamini & Hochberg correction. Batch and RUVr-computed factors were added in the design matrix. For the siEA vs siCTL 96 h DEG analysis, DTC1 and SU-CCS-1 samples were analyzed together utilizing three biological, and one technical replicate experiments of DTC1 for batch corrections.

Single cell RNAseq data from five adult healthy skin samples (dermis and epidermis) were downloaded from ZENODO [https://zenodo.org/record/4569496#.YMdnr5ozZYg]. Cells lacking a final clustering value were discarded, giving a total of 168,103 single cells for the analysis. The final clustering information from the original data was used to pool clusters according to their biological types and used for UMAP and box plot data visualization. Pooled clusters, with original cluster in parenthesis if deviating, were defined as follows: 01 Melanocyte, 02 Stromal Schwann (Schwann2), 03 Schwann (Schwann1), 04 Pericyte, 05 Pericyte inflammatory (Pericyte2), 06 Fibroblast, 07 Keratinocyte basal, 08 Keratinocyte suprabasal, 09 Lymphatic endothelium, 10 Vascular endothelium, 11T lymphocyte (Tc, Th, and Treg lymphocytes), 12 Plasma cell, 13 Innate lymphoid cell (ILC1_3, ILC2, ILC1_NK, NK), 14 Dermal APC (Inf_mono, moDC_3), 15 Epidermal APC, 16 Mast cell. The provided counts and cells annotations were analyzed using the Seurat package v 4.0.2 in R v 4.1.0, SeuratDisk v. 0.0.0.9019, zellkonverter v. 1.2.0, SeuratObject v.4.0.1. Boxplots were generated using the dittoSeq package v 1.4.1.

All other figures were generated using R v 4.1.0 (http://www.R-project.org/).

**Reporting summary**. Further information on research design is available in the Nature Research Reporting Summary linked to this article.

## Data availability
The raw data generated in this study are publicly available at the Gene Expression Omnibus (GEO) database under the series accession number GSE180198 which contains the datasets GSE180183 (ATAC-seq), GSE180187 (ChIP-seq), GSE180194 (HiChIP-seq) and GSE180196 (RNA-seq). The public human genome sequence data (GRCh37, hg19) used in this study is available through the Ensembl genome browser [https://grch37.ensembl.org/Homo_sapiens/Info/Index]. The public wt ATF1 ChIP-seq data used in this study are available through the ENCODE database [https://www.encodeproject.org/experiments/ENCSR253OON/, https://www.encodeproject.org/experiments/ENCSR091GVJ/]. The public single cell RNAseq data used in this study are available from ZENODO [https://zenodo.org/record/4569496#.YMdnr5ozZYg]. The remaining data are available within the Article, Supplementary Information or Source Data file.

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

## Acknowledgements

We thank Professor Fredrik Mertens, Division of Clinical Genetics, Department of Laboratory Medicine, Lund University, Lund, Sweden, for generously providing primary CCS tumor samples, and Professor T. Nielsen, Department of Pathology and Laboratory Medicine, University of British Columbia, Vancouver, Canada, for kindly providing the DTC1 cell line. We would like to thank Patricia Martin for technical support, and the UNIGE IGE3 Genomic platform for their support with high-throughput sequencing. This work was supported by Swiss National Science Foundation Professorship grant (PP00P3-157468/1 and PP00P3_183724), the Swiss Cancer League grant (KFS-3973-08-2016 and KFS4859-08-2019), the Fond'Action Contre le Cancer Foundation, the FORCE Foundation, the Liddy Shriver Early Career Research Award to (N.R.) and the Bertarelli Rare Cancers Fund (M.N.R.). E.M. and I.S. were supported by FNS Sinergia grant CRSII5-177266. M.N.R is supported by the Thomas F. and Diana L. Ryan MGH Research Scholar Award.

## Author contributions

E.M., M.N.R., G.B., I.S., and N.R. designed the study; E.M. performed the majority of the biochemistry and in vitro experiments; E.M., S.R., Shruti R., L.K.L, M.E.A, and B.N. generated and sequenced the NGS libraries; A.C. performed immunoprecipitation experiments; Y.H.X. performed beta-isox precipitation experiments; E.M. and C.F. designed and generated DNA expression plasmids; L.C.B. performed cell culture experiments; G.F, N.O., S.L.R and F.S. generated and analyzed the electron microscopy images; R.D., M.A. S.I, and V.P. performed bioinformatics analysis; I.L., I.C., G.P.N., A.D., D.S., G.C., and E.C. contributed to study conception; E.M., G.B., I.S., M.N.R. and N.R. analyzed the data and wrote the manuscript.

## Competing interests

M.N.R. receives support from ACD (Advanced Cell Diagnostics) and Merck-Serono for research unrelated to this study. The other authors declare no competing interests.
