## [Peer Review File · Nature Communications]

EWSR1-ATF1 dependent 3D connectivity regulates oncogenic and differentiation programs in Clear Cell Sarcoma.Reviewers' Comments:

Reviewer #1:

Remarks to the Author:

In this paper, Moller et al describes global DNA-binding properties of EWSR1-ATF1, a critical disease gene for human clear cell sarcoma (CCS). They showed distinct distributions of EWSR1-ATF1-binding peaks from those of wild type ATF1 using CCS cell lines and patient samples. Interestingly, EWSR1-ATF1 binding peaks in CCS were very different from those in angiomatoid fibrous histiocytoma, suggesting that EWSR1-ATF1 binding is affected by epigenetic background of the cell-of-origin. However, they also showed the evidence that wild type ATF1 interacted with EWSR1-ATF1 and was recruited by EWSR1-ATF1 on its DNA binding. They then silenced EWSR1-ATF1 and found reduction of H3K4me1 and H3K27Ac signals as well as ATAC-seq peaks around EWSR1-ATF1 binding sites. The recruitment of wild type ATF1 by EWSR1-ATF1 was important for chromatin activation as was indicated by increased binding of H3K27ac and p300 in EWSR1-ATF1-introduced HEK293 cells. Based on the previous study for EWSR1-FLI1 in Ewing sarcoma, the authors showed that the EWSR1 prion-like domain is required for the modulation of EWSR1-ATF1-specific DNA binding. The Hi-ChIP analysis to detect H3K27ac-associated chromatin loops suggests that EWSR1-ATF1 plays an important role in chromatin interactions around its target genes. Finally, they identified de novo direct targets of EWSR1-ATF1 by detecting modifications of DNA loops by EWSR1-ATF1 knockdown. The result was used for confirming the cell-of-origin of CCS as neural crests including melanocytes and Schwann cells with single cell RNA-seq public data.

This study is well prepared, and the paper is organized in a logical manner and well written. It will provide deeper insight into the pathogenesis of CCS and the functional role of EWSR1-ATF1, although the cell-of-origin of CCS has been substantially clarified by previous studies. The study will attract much interest of researchers not only in the sarcoma research field but also in the cancer biology and epigenetics. I would recommend the manuscript for publication if appropriate revision is made.

Specific comments:

1. The authors showed that there are significant differences in DNA binding site between EWSR1-ATF1 and wild type ATF1, and that this is reminiscent of the EWSR1-FLI1 in Ewing sarcoma. However, this is true only in part given the drastic change of the DNA binding motif from the canonical ETS motif to the GGAA microsatellite for EWSR1-FLI1. Please clarify whether binding motifs are different between EWSR1-ATF1 and wild type ATF1, and if they are same please explain the potential different activity of EWSR1 prion-like domain between CCS and Ewing sarcoma.
2. The authors define the wild type ATF1 binding sites in two different ways, i.e., ATF1 C-terminus binding sites without EWSR1 N-terminus binding (Fig. 1) and ATF1 N-terminus binding (Fig 3). How was the overlapping between these two potential wild type ATF1 sites? Please clarify this point.
3. The authors argued that EWSR1-ATF1 requires wt ATF1 recruitment for chromatin activation based on the co-occupancies of binding sites by EWSR1-ATF1, wt ATF1, ATAC, H3K27ac and p300. These data are insufficient to affirm the role of wt ATF1 by EWSR1-ATF1. I would request the authors to knockdown wt ATF1 and to show decrease of chromatin activation signals in the majority of EWSR1-ATF1 binding sites.
4. In Fig 5e, the authors indicated 3663 and 2569 direct target genes of EWSR1-ATF1 in SU-CCS1 and DTC1 cells, respectively. How were these direct target genes defined? I could not find the explanation for this point. In addition, the authors defined "direct target genes" by Hi-ChIP and EWSR1-ATF1 knockdown experiment in Fig 7c. Are these direct targets same as the previous ones in Fig. 5? Please clarify this point.

Reviewer #2:

Remarks to the Author:

The manuscript by Möller et al presents novel data and insights on the molecular mechanisms behind

clear cell sarcoma. This very rare tumor belongs to a large group of tumors caused by gene fusions involving the EWSR1 gene as donor of the 5' parts juxtaposed to a large number of alternative transcription factor partners. Two other, closely EWSR1 related genes, FUS and TAF15 may replace the EWSR1 as 5'-partner in some fusions. The common part for all these tumor types are thus the N-terminal parts of EWSR1 or FUS or TAF15 whereas the transcription factor partners vary. In this context, this study of the EWSR1/ATF1 fusion, adds a new level of insight to how the transcription factor partners contribute to the strong oncogenic activity of this large group of fusion oncogenes. The manuscript presents a wealth of data studying the genomic binding of EWSR1/ATF1 and the native ATF1 and overlaps with some activity associated chromatin marks. ATAC-seq is added to show open chromatin. And, adding data from Hi-ChIP, attempts to study chromatin-connections and loops are made.

The authors base their study on- CCS cell lines, but importantly, also CCS tumor tissues and tissue from another EWSR1/ATF1 caused entity. They also perform siRNA aided depletion of the fusion protein and transfect mesenchymal stem cells with the fusion oncogene.

The ChIPseq and ATAC-seq based studies are convincing with several data-sets showing overlapping localization of EWSR1/ATF1 with active chromatin marks. Data from experiments with siRNA depletion and forced expression of the fusion gene in stem cells show significant overlaps, thus supporting the conclusion that the fusion protein, in comparison with the normal ATF1, binds at novel genomic sites. Some that needs improvements:

1., The authors show an enrichment of normal ATF1 protein to the novel sites opened by the fusion protein and claim that this is an active recruitment by the fusion protein and an important part of the oncogenic mechanism. The ATF1 forms homo or heterodimers to bind to the cre-motif. ATF1 and several other cre-targeting potential dimerization partners could form dimers with the fusion protein allowing binding to cre sites. It is possible that the normal ATF1 proteins are recruited this way. But it may simply also be second hand enriched to the more open chromatin induced by the fusion protein. The data are not there to fully conclude that the fusion protein directly recruits the normal ATF1. This detail has to be changed or better explained. The fact that ATF1 forms dimers with EWSR1/ATF1 in coprecipitation experiments as shown here helps but there are other dimerization partners around in CCS cells.

2. The leucine zipper TFs generally binds DNA only as dimers. Many of the EWSR1/ATF1-binding sites were half-cre sites suggesting that the ATF1 part of the fusion-proteins (that maintain the leucine zipper dimerization parts) has formed DNA binding dimers with non-cre binding TFs that consequently defined the targeted DNA sites together with the ATF1 parts. These dimerization partners are thus of importance for understanding the recruitment of EWSR1/ATF1 proteins to new genomic sites and as some ATF1 compatible dimerization partners are cell type specific this could explain the cell type dependent outcome of a EWSR1/ATF1 fusion gene. Identification of potential ATF1 dimerization partners in CCS could therefore help explaining the dramatically divergent DNA binding. This could easily be done by looking expression of known ATF1 dimerization partners on mRNA level followed up by confirmation of hits on protein level.

3. The remarkably changed genomic localization of the fusion-protein compared to normal ATF1 is convincingly demonstrated in the manuscript but the possible mechanism behind this dramatic shift is not discussed. It could hardly be explained only by alternative dimerization partners to ATF1. Instead the strongly protein-protein interacting EWSR1-prion like N-terminal part has a determining role. The role of EWSR1 is experimentally addressed in the manuscript by testing a deletion mutant that disrupt the phase transitioning capacity of the fusion protein. However, while the results confirm the importance of the integrity of the EWSR1 it gives little on what causes the localization shift. In this context a discussion on the interaction of EWSR1 as well as the replacing FUS and TAF15 N-terminal parts with the SWI/SNF complexes is missing. The interactions are very strong and major parts of the fusion-proteins are bound to SWI/SNF (Lindén et al doi: 10.15252/embr.201845766). (Boulay et al 2017 doi: 10.1016/j.cell.2017.07.036 already cited). A discussion and analysis of how the N-terminal prion like parts and the various transcription factor partners act together and contribute to the large genomic changes would improve the manuscript!

4. The Hi-ChIP data suggest very large changes in chromatin looping and structure! In Ewing sarcoma, myxoid liposarcoma and desmoplastic small round cell tumors containing similar fusion oncogenes, the

number of fusion protein copies are small and it is hard to envision how they cause such dramatic changes. A short discussion on this would be interesting!

Reviewer #3:

Remarks to the Author:

Möller et al. report that EWSR1-ATF1 dependent 3D connectivity regulates oncogenic and differentiation programs in Clear Cell Sarcoma. This is an interesting work shedding light on tumorigenic effects of the fusion protein. Some issues remain.

WT ATF1 seems, perhaps a bit unexpectedly, to be one key partner of the fusion protein. Is ATF1 overexpressed in CCS? One would presume yes if it is highly relevant for tumorigenesis, though of course one allele is sacrificed for the fusion.

While the story looks conceivable more information on the quality of the ChIP-seq experiments would be important. Authors could easily use e.g. deeptools' multiBigwigSummary bins -function combined with plotCorrelation to give unbiased view of the similarity/dissimilarity of ChIP profiles. Analysis of concordancy and discordancy of ChIP-seq replicates (technical, and biological i.e. cell lines vs primary tumor) should be presented. Unfortunately the authors have not even filled in the Reporting Summary here (ChIP-Seq>Methodology>Replicates: "Describe the experimental replicates, specifying number, type and replicate agreement").

"Chimeric proteins resulting from chromosomal translocations play a major role as driver oncogenes in cancer."

Proteins are not oncogenes, some wording needed.

"CCS tumor"

CCS is a tumor so please just use CCS.

Please define "distal region" in main text

"The majority of EWSR1-ATF1 peaks (84%) were associated with distal regions and the remaining 16% of peaks were located at gene promoters (TSS) (Fig. 1a)."

No peaks anywhere else; please explain.

"Moreover, distal sites bound by EWSR1-ATF1 are less frequently observed in other cell types since they display lower average DNase I signals in 113 human cell types profiled by ENCODE (GSE29692)20, compared to wt ATF1 sites (Fig. 1b). "

Perhaps I cannot follow the thoughts here but this evidence seems quite vague.

"Similarly, EWSR1-ATF1 binding sites at TSSs were associated with H3K4me3, H3K27ac and ATAC signals (Fig. 1c), consistent with the notion that the fusion protein behaves as a transcriptional activator at both regulatory regions. Notably, the three primary tumors CCS1-CCS3, all of which expressed EWSR1-ATF1, displayed similar chromatin state profiles at EWSR1-ATF1 peaks (Fig. 1c)."

No indication here about the quality of chromatin state profiles; e.g. if H3K4me1 and H3K4me3 signals were randomly scattered throughout the genome this would give similar results.

"Our observations therefore suggest that EWSR1-ATF1 action may be conditioned by properties intrinsic to the tumor and its cell of origin."

True, though this was clear already because the fusion is not even selected for in most tissues.

"To investigate the potential cooperativity between the fusion protein and NCSC-related TFs, we profiled the TFAP2A binding sites by ChIP-seq in the two CCS cell lines and observed TFAP2A to be present at the majority of the 2385 EWSR1-ATF1 bound sites in both (59% in DTC1 and 82% in SU-

CCS-1, Extended data Fig. 1e)."

Again, this result is not entirely convincing without information on the quality of TFAP2A ChIP since only EWSR1-ATF1 bound sites are shown.

"Overall, the two cell lines responded similarly to EWSR1-ATF1 depletion with a significant decrease in enhancer activity, as illustrated by the reduction in both H3K4me1 and H3K27ac signals at distal binding sites (Fig. 2c middle), as well as a reduction in H3K4me3 and H3K27ac signals at TSS sites (Extended data Fig. 2b)."

To believe this, the reader would need to see that the ChIP signal enrichment at all TSS and/or FRiP are somewhat similar throughout the genome with and without the depletion for the chromatin marks because the reduction at certain binding sites might only reflect less successful immunoprecipitation and not biological difference. Extended Data Fig 2c could somewhat answer the question but it is not clear how many peaks are unchanged. The figure legend is also unclear "c, Scatter plots of H3K27ac peak FC scores ($\log_2(\text{siEA}) - \log_2(\text{siCTL})$) in siEA- and siCTL-treated DTC1 and SU-CCS-1 cells. At a FC 2 cutoff, 617 and 5872 peaks were increased in siEA-treated DTC1 and SU-CCS-1, respectively (blue); whereas 2209 and 4942, respectively, were decreased (red)." Perhaps FC scores are not plotted but $\log_2(\text{siEA})$ and $\log_2(\text{siCTL})$, but what exactly are these values is not explained. Based on the figure, there seems to be very low correlation between SUCCS1 siEA and SUCCS1 siCTL genomewide indicating possible problems in ChIP.

"Overall, the two cell lines responded similarly to EWSR1-ATF1 depletion with a significant decrease in enhancer activity, as illustrated by the reduction in both H3K4me1 and H3K27ac signals at distal binding sites (Fig. 2c middle), as well as a reduction in H3K4me3 and H3K27ac signals at TSS sites (Extended data Fig. 2b)."

Was this phenomenon specifically seen at the EWSR1-ATF1 binding sites?

"genome wide H3K27ac signals showed both decreases and increases in EWSR1-ATF1-depleted DTC1 and SU-CCS-1 tumor cells (Extended data Fig. 2c), attesting to profound changes in their global epigenetic landscape."

How much could this be due to variation between replicates?

"The observed decrease in the ATAC signal also suggests that EWSR1-ATF1 may be involved in generating de novo distal regulatory elements through pioneering properties similar to the ones identified for EWSR1-FLI1 in Ews2."

How do the authors deduct the pioneering function from these data?

"Wt ATF1 was found to be present at the majority of the EWSR1-ATF1 binding sites in both cell lines (62.5% in DTC1, 75% in SU-CCS-1, Fig. 3a), and confirmed in the primary CCS1 tumor, although at a lower percentage of EWSR1-ATF1 sites (38%, Fig. 3a), a difference that may be related to the technical challenge of profiling TFs in frozen tumor samples."

Here the authors correctly acknowledge that ChIP quality issues may affect the results.

"Consistent with our previous results, the binding pattern of EWSR1-ATF1 remained predominantly restricted to distal genomic regions (7% of TSSs), whereas the YS37 mutant displayed a significant increase in promoter binding (56% of TSS), close to the binding pattern of wt ATF1 (60% of TSS) (Fig. 4e)."

Genome-wide correlation between TFs' ChIP-seq profiles would be much more informative.

"These observations support the notion that CCS may originate from precursor cells along the neural crest lineage, and suggest that the fusion protein may induce a cellular reprogramming toward a more undifferentiated state by retargeting the lineage-specifying TFAP2A, SOX10 and MITF TFs from their original binding sites."

The authors have profiled the TFAP2A binding sites by ChIP-seq in DTC1 and SU-CCS-1 cells so it would be useful to see the change in the binding sites upon EWSR1-ATF1 removal to confirm this suggestion

"that EWSR1-ATF1 may display similar neomorphic pioneer properties that enable tumor initiation in CCS."

Are these fusions strongly clonal in CCS (like they should be if tumor initiating)?

"Thus, the collaborative TF network identified in this study may serve EWSR1-ATF1 to induce the oncogenic reprogramming required for tumor development."

The collaborative TF network proposed makes sense and it is conceivable that the tumor needs to work with factors present in the cell of origin; where there any other factors enriched?

"CPM-normalized BigWig files were generated using bamCoverage"

Are the reads extended to fragment size? They should be for visualization. The tools' documentation also says: By default, the read length is not extended. This is the preferred setting for spliced-read data like RNA-seq, where one usually wants to rely on the detected read locations only. A read extension would neglect potential splice sites in the unmapped part of the fragment. Other data, e.g. Chip-seq, where fragments are known to map contiguously, should be processed with read extension (--extendReads [INTEGER]).

Response to Reviewer's comments.

We would like to thank the 3 reviewers for their comments and suggestions, which we believe helped us to improve the quality of our study. We now provide a detailed point-by-point response to their questions, which we hope the reviewers will find satisfactory.

Reviewer #1, expert in EWS/clear cell sarcoma/functional genomics (Remarks to the Author):

In this paper, Moller et al describes global DNA-binding properties of EWSR1-ATF1, a critical disease gene for human clear cell sarcoma (CCS). They showed distinct distributions of EWSR1-ATF1-binding peaks from those of wild type ATF1 using CCS cell lines and patient samples. Interestingly, EWSR1-ATF1 binding peaks in CCS were very different from those in angiomatoid fibrous histiocytoma, suggesting that EWSR1-ATF1 binding is affected by epigenetic background of the cell-of-origin. However, they also showed the evidence that wild type ATF1 interacted with EWSR1-ATF1 and was recruited by EWSR1-ATF1 on its DNA binding. They then silenced EWSR1-ATF1 and found reduction of H3K4me1 and H3K27Ac signals as well as ATAC-seq peaks around EWSR1-ATF1 binding sites. The recruitment of wild type ATF1 by EWSR1-ATF1 was important for chromatin activation as was indicated by increased binding of H3K27ac and p300 in EWSR1-ATF1-introduced HEK293 cells. Based on the previous study for EWSR1-FLI1 in Ewing sarcoma, the authors showed that the EWSR1 prion-like domain is required for the modulation of EWSR1-ATF1-specific DNA binding. The Hi-ChIP analysis to detect H3K27ac-associated chromatin loops suggests that EWSR1-ATF1 plays an important role in chromatin interactions around its target genes. Finally, they identified de novo direct targets of EWSR1-ATF1 by detecting modifications of DNA loops by EWSR1-ATF1 knockdown. The result was used for confirming the cell-of-origin of CCS as neural crests including melanocytes and Schwann cells with single cell RNA-seq public data.

This study is well prepared, and the paper is organized in a logical manner and well written. It will provide deeper insight into the pathogenesis of CCS and the functional role of EWSR1-ATF1, although the cell-of-origin of CCS has been substantially clarified by previous studies. The study will attract much interest of researchers not only in the sarcoma research field but also in the cancer biology and epigenetics. I would recommend the manuscript for publication if appropriate revision is made.

Specific comments:

1. The authors showed that there are significant differences in DNA binding site between EWSR1-ATF1 and wild type ATF1, and that this is reminiscent of the EWSR1-FLI1 in Ewing sarcoma. However, this is true only in part given the drastic change of the DNA binding motif from the canonical ETS motif to the GGAA microsatellite for EWSR1-FLI1. Please clarify whether binding motifs are different between EWSR1-ATF1 and wild type ATF1, and if they are same please explain the potential different activity of EWSR1 prion-like domain between CCS and Ewing sarcoma.

Response: As correctly pointed out by the reviewer, the situation in EwS and CCS is different in that in CCS EWSR1-ATF1 and wt ATF1 indeed recognize the same DNA binding motif, whereas in EwS the EWSR1-FLI1 fusion protein is able to recognize both canonical single GGAA ETS binding sites, as well as related GGAA repeat sequences. Although we expect the prion-like domain of EWSR1-FLI1 and EWSR1-ATF1 to provide these two fusion proteins with similar functions (i.e. the acquisition of pioneering properties through the recruitment of wt EWSR1 and SWI/SNF remodeling complexes), this would imply the presence of repetitive sequences of the wt ATF1 motif in the human genome, similar to the GGAA microsatellite repeats. To evaluate the genomic co-occurrence of ATF1 motifs, as compared to FLI1 motifs, we applied the following strategy;

1. Scan the genome for the ATF1 consensus motif and the ATF1 half-site (motif matrices as identified by HOMER), respectively, using PWMScan (<https://ccg.epfl.ch/pwmtools/pwmtools.php>). Motif positions are saved as a bed file.
2. Scan the genome for the FLI1 consensus motif and the FLI1 short motifs (motif matrices as identified in the HOCOMOCO v100 Human TF collection, FLI1_HUMAN.H11MO.0.A), respectively, as above.
3. Perform an auto-correlation genomic analysis of each motif bed file using ChIP-Cor (https://ccg.epfl.ch/chipseq/chip_cor.php). This tool calculates the occurrence frequency of close motifs.

The results of this analysis are shown in Response-to-Reviewers (**RtR**) **Fig 1a**, and clearly indicate that repetitive sequence of wt ATF1 motifs almost never occur in the human genome, as compared to the FLI1 motifs. This may therefore explain the lack of drastic changes in DNA binding motif for EWSR1-ATF1, as compared to EWSR1-FLI1.

2. The authors define the wild type ATF1 binding sites in two different ways, i.e., ATF1 C-terminus binding sites without EWSR1 N-terminus binding (Fig. 1) and ATF1 N-terminus binding (Fig 3). How was the overlapping between these two potential wild type ATF1 sites? Please clarify this point.

Response: We apologize for this confusion. Since distinct ChIP-seq antibodies targeting the same protein may differ in their sensitivity and specificity, we chose to base our definition of EWSR1-ATF1 and wt ATF1 peak categories starting from the same ATF1 C and EWSR1 N ChIP-seq data. The ATF1 N ChIP-seq was performed subsequently as an additional validation step. To evaluate the overlap, we have now looked for ATF1 N signal at the 1288 wt ATF1 peak set (defined as ATF1 C-pos / EWSR1-neg sites), and found ATF1 N signal to be present at these sites in both ATF1 N ChIP-seq replicates of DTC1 and SU-CCS-1 cells. These results are presented in **RtR Fig 1b**.

3. The authors argued that EWSR1-ATF requires wt ATF1 recruitment for chromatin activation based on the co-occupancies of binding sites by EWSR1-ATF1, wt ATF1, ATAC, H3K27ac and p300. These data are insufficient to affirm the role of wt ATF1 by EWSR1-ATF1. I would request the authors to knockdown wt ATF1 and to show decrease of chromatin activation signals in the majority of EWSR1-ATF1 binding sites.

Response: We fully agree with the reviewer that wt ATF1 depletion would be informative in this context, and we have therefore added this experiment to the main results (wt ATF1 depletion in both CCS cell lines using an shRNA hairpin specifically targeting the 5' sequence of the ATF1 transcript, which is absent in EWSR1-ATF1). We experienced some issues with this experiment, because a complete or prolonged wt ATF1 knock-down caused the cells to undergo apoptosis, preventing us to collect enough viable cells for downstream analyses. To be able to perform this experiment, we therefore decided to generate CCS cells with a partial loss of wt ATF1 expression, and to analyze them at 72 hours post-lentiviral infection. Despite the suboptimal knock-down efficacy, we were still able to observe a decrease in H3K27ac signal at EWSR1-ATF1/wt ATF1 binding sites in depleted cells, supporting the role of wt ATF1 in chromatin activation. In agreement with this, q-PCR analysis for a panel of genes whose regulatory elements show EWSR1-ATF1 and wt ATF1 co-binding in CCS lines, showed a decrease in their expression level upon partial wt ATF1 depletion. These data are now presented in the revised version of the manuscript and showed in Fig. 3f and Supplementary Fig. 3.

4. In Fig 5e, the authors indicated 3663 and 2569 direct target genes of EWSR1-ATF1 in SU-CCS1 and DTC1 cells, respectively. How were these direct target genes defined? I could not find the explanation for this point. In addition, the authors defined "direct target genes" by Hi-ChIP and EWSR1-ATF1 knockdown experiment in Fig 7c. Are these direct targets same as the previous ones in Fig. 5? Please clarify this point.

Response: We apologize and agree with the reviewer for the lack of clarity about this analysis. The definition of the EWSR1-ATF1 direct target genes shown in Fig. 5e is based on the association of their TSSs to at least one EWSR1-ATF1 binding site, either directly or through H3K27ac-associated enhancer elements, as inferred by the 3D chromatin looping profiles obtained by Hi-ChIP. We have now specified this definition in the Methods section of the revised manuscript. Importantly, the expression of these direct target genes is reduced upon EWSR1-ATF1 depletion, as shown in Fig. 6a and b.

On the contrary, the genes shown in Fig. 7 are not EWSR1-ATF1 direct target genes. In Fig. 7, we explored the notion that following EWSR1-ATF1 depletion, CCS tumor cells may revert to a differentiation state and transcriptional program reminiscent of their putative cell-of-origin, and hypothesized that this feature may be induced by the emergence of lineage-specific enhancer elements. To validate this hypothesis, we first identified H3K27ac-associated enhancers that were generated *de novo* in siEA CCS cells using ChIP-seq data (as detailed in the Method section). We then used the corresponding Hi-ChIP looping profiles to identify the direct target genes of these *de novo* enhancer elements, that we thereby named "*de novo direct targets*". As the H3K27ac signal at these sites was increased in siEA cells, the expression of their corresponding direct target genes also showed an increase upon EWSR1-ATF1 removal (Fig. 7c).

Reviewer #2, expert in RNA-seq/ChIP-seq (Remarks to the Author):

The manuscript by Möller et al presents novel data and insights on the molecular mechanisms behind clear cell sarcoma. This very rare tumor belongs to a large group of tumors caused by gene fusions involving the EWSR1 gene as donor of the 5' parts juxtaposed to a large number of alternative transcription factor partners. Two other, closely EWSR1 related genes, FUS and TAF15 may replace the EWSR1 as 5'-partner in some fusions. The common part for all these tumor types are thus the N-terminal parts of EWSR1 or FUS or TAF15 whereas the transcription factor partners vary. In this context, this study of the EWSR1/ATF1 fusion, adds a new level of insight to how the transcription factor partners contribute to the strong oncogenic activity of this large group of fusion oncogenes. The manuscript presents a wealth of data studying the genomic binding of EWSR1/ATF1 and the native ATF1 and overlaps with some activity associated chromatin marks. ATAC-seq is added to show open chromatin. And, adding data from HI-ChIP, attempts to study chromatin-connections and loops are made. The authors base their study on CCS cell lines, but importantly, also CCS tumor tissues and tissue from another EWSR1/ATF1 caused entity. The authors also perform siRNA aided depletion of the fusion protein and transfect mesenchymal stem cells with the fusion oncogene. The ChIPseq and ATAC-seq based studies are convincing with several data-sets showing overlapping localization of EWSR1/ATF1 with active chromatin marks. Data from experiments with siRNA depletion and forced expression of the fusion gene in stem cells show significant overlaps, thus supporting the conclusion that the fusion protein, in comparison with the normal ATF1, binds at novel genomic sites. Some that needs improvements:

1. The authors show an enrichment of normal ATF1 protein to the novel sites opened by the fusion protein and claim that this is an active recruitment by the fusion protein and an important part of the oncogenic mechanism. The ATF1 forms homo or heterodimers to bind to the cre-motif. ATF1 and several other cre-targeting potential dimerization partners could form dimers with the fusion protein allowing binding to cre sites. It is possible that the normal ATF1 proteins are recruited this way. But it may simply also be second hand enriched to the more open chromatin induced by the fusion protein. The data are not there to fully conclude that the fusion protein directly recruits the normal ATF1. This detail has to be changed or better explained. The fact that ATF1 forms dimers with EWSR1/ATF1 in coprecipitation experiments as shown here helps but there are other dimerization partners around in CCS cells.

Response: We fully agree with the reviewer about the possibility that other CRE-motif binding TFs may participate in EWSR1-ATF1-induced chromatin activation, and we have now added this wording of caution to the revised text accordingly. The reviewer also raises a valid point that normal ATF1 proteins may be recruited to EWSR1-ATF1 binding sites following the opening of these genomic regions by the fusion protein. This is a very challenging point, which we don't believe can be addressed by any specific experiment. However, our results showing a loss of wt ATF1 binding upon EWSR1-ATF1 depletion (Fig 3b), but not of EWSR1-ATF1 binding following wt ATF1 removal (Supplementary Fig. 3) speak in favor of a potential direct recruitment of wt ATF1 by the fusion protein. We have also added a new co-immunoprecipitation experiment showing the interaction between EWSR1-ATF1 and wt ATF1 in the DTC1 cell line, conforming our previous results obtained in 293T cells (Fig. 3c).

2. The leucine zipper TFs generally binds DNA only as dimers. Many of the EWSR1/ATF1-binding sites were half-cre sites suggesting that the ATF1 part of the fusion-proteins (that maintain the leucine zipper dimerization parts) has formed DNA binding dimers with non-cre binding TFs that consequently defined the targeted DNA sites together with the ATF1 parts. These dimerization partners are thus of importance for understanding the recruitment of EWSR1/ATF1 proteins to new genomic sites and as some ATF1 compatible dimerization partners are cell type specific this could explain the cell type dependent outcome of a EWSR1/ATF1 fusion gene. Identification of potential ATF1 dimerization partners in CCS could therefore help explaining the dramatically divergent DNA binding. This could easily be done by looking expression of known ATF1 dimerization partners on mRNA level followed up by confirmation of hits on protein level.

Response: We agree with this insight and recognize that non-CRE binding dimerization partners of wt ATF1 could play an important role in modulating the chromatin binding profile and transcriptional activity of EWSR1-ATF1. Given that our motif enrichment analysis of EWSR1-ATF1 binding sites highlighted TFAP and SOX motifs, suggesting that these TFs may be important co-factors, and since TFAP2A and SOX10 are recognized as master TFs in both normal melanocytes and melanoma tumor cells, we have now included ChIP-seq profiles for SOX10 in wt DTC1 and SU-CCS-1 cells and confirmed the presence of this TF at a majority of EWSR1-ATF1 binding sites. Moreover, we also generated the genomic occupancy profiles for MITF, given that this TF has been previously identified as a major player in CCS, as well as a member of a collaborative TF network regulating melanocytic differentiation, along with TFAP2A and SOX10. Consistent with these notions, MITF was also found to be present at a majority of EWSR1-ATF1 bound sites. We also profiled TFAP2A, SOX10 and MITF binding sites in EWSR1-ATF1-depleted SU-CCS-1 cells, and demonstrated that these TFs are displaced from EWSR1-ATF1 binding sites upon removal of the fusion protein. In aggregate, our new results identify a network of TFs associated with EWSR1-ATF1, and whose chromatin occupancy partially depends on the presence of the fusion protein. These new results are now shown in Supplementary Fig. 1g and 2d, and highlighted in the corresponding sections of our revised manuscript.

3. The remarkably changed genomic localization of the fusion-protein compared to normal ATF1 is convincingly demonstrated in the manuscript but the possible mechanism behind this dramatic shift is not discussed. It could hardly be explained only by alternative dimerization partners to ATF1. Instead the strongly protein-protein interacting EWSR1-prion like N-terminal part has a determining role. The role of EWSR1 is experimentally addressed in the manuscript by testing a deletion mutant that disrupt the phase transitioning capacity of the fusion protein. However, while the results confirm the importance of the integrity of the EWSR1 it gives little on what causes the localization shift. In this context a discussion on the interaction of EWSR1 as well as the replacing FUS and TAF15 N-terminal parts with the SWI/SNF complexes is missing. The interactions are very strong and major parts of the fusion-proteins are bound to SWI/SNF (Lindén et al doi: 10.15252/embr.201845766). (Boulay et al 2017 doi: 10.1016/j.cell.2017.07.036 already cited). A discussion and analysis of how the N-terminal prion like parts and the various transcription factor partners act together and contribute to the large genomic changes would improve the manuscript!

Response: We fully agree with the reviewer about the important role of EWSR1 N-terminal domain interaction with the SWI/SNF complex in determining the chromatin binding properties displayed by EWSR1-ATF1. To address this point, we now provide BRG1 co-IP studies in the result section (Supplementary Fig. 1c), showing that the interaction with BRG1 is a property limited to EWSR1-ATF1, and not shared with wt ATF1. This suggests that the ability of EWSR1-ATF1 to recruits the SWI/SNF remodeling complex is a neo-morphic property acquired by the fusion protein, most probably underlying the different chromatin binding profiles observed between the EWSR1-ATF1 and wt ATF1 TFs. These new results are now presented and discussed in the revised version of our manuscript.

4.The Hi-ChIP data suggest very large changes in chromatin looping and structure! In Ewing sarcoma, myxoid liposarcoma and desmoplastic small round cell tumors containing similar fusion oncogenes, the number of fusion protein copies are small and it is hard to envision how they cause such dramatic changes. A short discussion on this would be interesting!

Response: We agree with the reviewer that the changes in 3D cis-connectivity following EWSR1-ATF1 depletion are very dramatic, despite the limited number of binding sites identified for the fusion protein. However, we would like to underscore the notion that EWSR1-ATF1 has been shown to represent the major driver event in the development of CCS, and is therefore plausible that the process of tumor initiation may require major changes in chromatin organization, in order to provide permissive oncogenic transcriptional programs. In keeping with this, in our study we observe changes in 3D looping and gene expression that are either directly linked to EWSR1-ATF1 activity, or resulting from major alterations in tumor cells differentiation. Gaining the ability to profoundly impact chromatin interactions may even represent a potential key feature of sarcoma-promoting fusion proteins like EWSR1-FLI1, EWSR1-ATF1 and EWSR1-WT1. We are now discussing this interesting point in the revised version of the manuscript.

Reviewer #3, expert in RNA-seq/ChIP-seq/ATAC-seq/HiChIP (Remarks to the Author):

Möller et al. report that EWSR1-ATF1 dependent 3D connectivity regulates oncogenic and differentiation programs in Clear Cell Sarcoma. This is an interesting work shedding light on tumorigenic effects of the fusion protein. Some issues remain.

1. WT ATF1 seems, perhaps a bit unexpectedly, to be one key partner of the fusion protein. Is ATF1 overexpressed in CCS? One would presume yes if it is highly relevant for tumorigenesis, though of course one allele is sacrificed for the fusion.

Response: We understand the potential interest of assessing the expression levels of wt ATF1. Following the reviewer's suggestion, we have surveyed wt ATF1 expression across a panel of 14 different sarcoma cell lines profiled by the CCLE consortium (<https://sites.broadinstitute.org/ccle/>). We have normalized the expression values by the median of all alive genes scores (scores being the log₂ of RSEM reads counts) for each tumor type, and compared this to the wt ATF1 expression level in primary CCS and AFH tumors, as well as the two CCS cell lines included in the present study. For our primary samples and cell lines, we also looked at the reads counts that align uniquely to the 5' mRNA sequence of ATF1, which is not included in EWSR1-ATF1. Our results show that wt ATF1 expression does not differ significantly across all tumor cell lines analyzed, and in particular does not appear to be over expressed in CCS. These results are shown in **RtR_Fig1c**.

Our results are in agreement with other studies showing that a gene do not necessarily need to be over expressed in order to be highly relevant for tumorigenesis. This notion is particularly relevant for TFs, whose combinatorial activity tends to be more important than the expression level of each single member of the network. In keeping with this, there are numerous examples of genes being relevant for tumorigenesis without any change in expression. ATF1 itself has been reported to drive tumorigenesis by different mechanisms, including overexpression, suppression or enhanced activity. We are now addressing this point in the revised version of the manuscript¹⁻⁴.

2. While the story looks conceivable more information on the quality of the ChIP-seq experiments would be important. Authors could easily use e.g. `deeptools' multiBigwigSummary bins -function combined with plotCorrelation` to give unbiased view of the similarity/dissimilarity of ChIP profiles. Analysis of concordancy and discordancy of ChIP-seq replicates (technical, and biological i.e. cell lines vs primary tumor) should be presented. Unfortunately the authors have not even filled in the Reporting Summary here (ChIP-Seq>Methodology>Replicates: "Describe the experimental replicates, specifying number, type and replicate agreement").

Response: We acknowledge that a part of the Reporting summary (ChIP-Seq>Methodology>Replicates) was missing. This was an error from our side, and we thank the reviewer for bringing this issue to our attention. We have now added the missing part to the Reporting summary accordingly. In the Methods section of our manuscript (Methods/TF and histone mark ChIP-seq data analysis) it is also explained that

two replicates each for the DTC1 and SU-CCS-1 cell lines were used to generate the EWSR1-ATF1 reference peak set.

We certainly agree that ChIP-seq quality evaluation is very important, and we have performed this type of quality control on all our data prior to any downstream analysis. Following the reviewer's suggestion, we have now performed Pearson correlation analysis of our ChIP-seq profiles, including cell line replicates and primary samples (**RtR_Fig2**). For the DTC1 and SU-CCS-1 cell lines replicates, the correlation is > 0.9 for all antibodies, and > 0.97 for the majority of them, consistent with a very high quality of each ChIP replicate experiment. When comparing the different cell line experiments to each other, the correlation is also very significant (> 0.6). Finally, when comparing cell lines and primary samples, we identify a good correlation for most of them (> 0.6). A limited number of H3K27ac ChIP-seq profiles shows a slightly lower correlation coefficient, mostly related to the differences between cultured cells and tissues ChIP-seq experiments. However, all correlations show a p-value of $< 2.2e-16$, corresponding to the lower p-value limit provided by the Pearson correlation calculation in R.

3. "Chimeric proteins resulting from chromosomal translocations play a major role as driver oncogenes in cancer." Proteins are not oncogenes, some wording needed.

Response: We agree that proteins are not oncogenes, therefore the sentence has been changed to: "Chimeric proteins resulting from chromosomal translocations play a major role as driver oncogenic events in cancer".

4. "CCS tumor". CCS is a tumor so please just use CCS.

Response: While we agree that the abbreviation CCS does indeed already refer to a tumor, we also feel that it wouldn't be accurate to only use the CCS abbreviation in our manuscript. In the text, the abbreviation CCS is used not only in the context of "CCS tumor" but also for "CCS tumor cells" and "CCS tumor samples". Substituting "CCS tumor" with "CCS" would require us to write "CCS cells" and "CCS samples" which would cause confusion as to which material is referred to. We will therefore feel more comfortable keeping the terminology "CCS tumor", and we hope the reviewer will agree with this strategy.

5. Please define "distal region" in main text "The majority of EWSR1-ATF1 peaks (84%) were associated with distal regions and the remaining 16% of peaks were located at gene promoters (TSS) (Fig. 1a)." No peaks anywhere else; please explain.

Response: The definition "distal region" is already explained in Methods as "The peaks within +/- 1 kb of a known gene's TSS (ucsc_RefFlat_07_08_2016) were labelled as TSS. The remnant peaks were evaluated for H3K4me3 enrichment as above, and labelled as TSS if H3K4me3 enriched regions were identified in four ChIP-seq experiments, giving in total 374 TSS-associated EWSR1-ATF1 peaks. All the other peaks were

labelled as Distal (n=2011). For the genomic distributions, the Distal peaks were annotated as intragenic or intergenic according to the ucsc_RefFlat_07_08_2016 list". It is also clear from this description that there are no other peaks located anywhere else in the genome. ChIPseq peaks are indeed categorized as either TSS or Distal and we are not aware of additional genomic location categories that should be included.

6. "Moreover, distal sites bound by EWSR1-ATF1 are less frequently observed in other cell types since they display lower average DNase I signals in 113 human cell types profiled by ENCODE (GSE29692)20, compared to wt ATF1 sites (Fig. 1b)." Perhaps I cannot follow the thoughts here but this evidence seems quite vague.

Response: Our analysis in Fig1b shows that the DNase I signal profiles for the binding sites of EWSR1-ATF1 and wt ATF1 are clearly different across a panel of human tissues and cell types, with markedly lower signal for the fusion protein bound sites. The lower DNase I signal detected at EWSR1-ATF1 binding sites indicates that the majority of those regions display decreased accessibility across a variety of cell types. This observation suggests that EWSR1-ATF1 binding participates in converting these otherwise inaccessible sites into active enhancer elements, a process which could underlie the differences in genomic occupancy between EWSR1-ATF1 and wt ATF1. Based on these results, and the ability of EWSR1-ATF1 to generate de novo enhancers in MSCs, in our study we propose the possibility that EWSR1-ATF1 may act as a pioneer transcription factor which binds a distinct set of distal elements in CCS.

7. "Similarly, EWSR1-ATF1 binding sites at TSSs were associated with H3K4me3, H3K27ac and ATAC signals (Fig. 1c), consistent with the notion that the fusion protein behaves as a transcriptional activator at both regulatory regions. Notably, the three primary tumors CCS1-CCS3, all of which expressed EWSR1-ATF1, displayed similar chromatin state profiles at EWSR1-ATF1 peaks (Fig. 1c)." No indication here about the quality of chromatin state profiles; e.g. if H3K4me1 and H3K4me3 signals were randomly scattered throughout the genome this would give similar results.

Response: We understand the concern about the quality of chromatin histone marks profiles. To show that the association of EWSR1-ATF1 binding with an active chromatin state is not the result of the histone mark signals being randomly scattered across the genome, we applied the following strategy to plot H3K4me1, H3K27ac, H3K4me3 and ATAC signals at random genomic regions:

All ChIP-seq peaks generated from all wt cell lines experiments were pooled and regions within 10kb merged. The regions that did not overlap with any of those peaks were then extracted, yielding 7870 TSSs and 1'824'990 enhancers (DENdb list⁵). A similar number of these TSSs (n=1967) and enhancers (n=2382) were randomly picked for heat map generation.

It is evident from the heatmaps that the H3K4me1, H3K27ac and ATAC signals are very weak, or non-existent, at these random genomic regions (**RtR_Fig3a**). Only H3K4me3 signal is present at some of the TSSs in the primary CCS tumor samples, however at much lower intensity (color scale max 0.32) as compared to the signal at EWSR1-ATF1 bound sites (Fig. 1c, color scale max 2.5). Thus, the association of EWSR1-ATF1 binding with the presence of these chromatin mark signals is not due to a random distribution of their signal.

“Our observations therefore suggest that EWSR1-ATF1 action may be conditioned by properties intrinsic to the tumor and its cell of origin.” True, though this was already clear because the fusion is not even selected for in most tissues.

8. "To investigate the potential cooperativity between the fusion protein and NCSC-related TFs, we profiled the TFAP2A binding sites by ChIP-seq in the two CCS cell lines and observed TFAP2A to be present at the majority of the 2385 EWSR1-ATF1 bound sites in both (59% in DTC1 and 82% in SU-CCS-1, Extended data Fig. 1e)." Again, this result is not entirely convincing without information on the quality of TFAP2A ChIP since only EWSR1-ATF1 bound sites are shown.

Response: We again acknowledge the importance of the TFAP2A ChIP-seq quality. To illustrate this, we generated a plot depicting the TFAP2A ChIP-seq signals across all genomic TSSs (ucsc_RefFlat_07_08_2016), and which shows strong signal at a majority of sites (**RtR_Fig3b**, left). To show that the TFAP2A signal is not scattered randomly in the genome, we plot the TFAP2A signal at the same random genomic sites described under point 7 above (**RtR_Fig3b**, right). Accordingly, we do not think that the presence of TFAP2A at EWSR1-ATF1 binding sites is due to random localization of TFAP2A signal in the genome.

9. "Overall, the two cell lines responded similarly to EWSR1-ATF1 depletion with a significant decrease in enhancer activity, as illustrated by the reduction in both H3K4me1 and H3K27ac signals at distal binding sites (Fig. 2c middle), as well as a reduction in H3K4me3 and H3K27ac signals at TSS sites (Extended data Fig. 2b)." To believe this, the reader would need to see that the ChIP signal enrichment at all TSS and/or FRiP are somewhat similar throughout the genome with and without the depletion for the chromatin marks because the reduction at certain binding sites might only reflect less successful immunoprecipitation and not biological difference. Extended Data Fig 2c could somewhat answer the question but it is not clear how many peaks are unchanged. The figure legend is also unclear "c, Scatter plots of H3K27ac peak FC scores ($\log_2(\text{siEA}) - \log_2(\text{siCTL})$) in siEA- and siCTL-treated DTC1 and SU-CCS-1 cells. At a FC 2 cutoff, 617 and 5872 peaks were increased in siEA-treated DTC1 and SU-CCS-1, respectively (blue); whereas 2209 and 4942, respectively, were decreased (red)." Perhaps FC scores are not plotted but $\log_2(\text{siEA})$ and $\log_2(\text{siCTL})$, but what exactly are these values is not explained. Based on the figure, there seems to be very low correlation between SUCCS1 siEA and SUCCS1 siCTL genomewide indicating possible problems in ChIP.

Response: The reviewer raises the concern about ChIP-seq quality differences between the siCTL and siEA conditions, as a potential reason explaining the lower histone mark signal identified in siEA CCS cells. We fully agree that this is a valid question, and to address this point we generated heatmaps of H3K27ac and ATAC signals at all TSSs genome wide (ucsc_RefFlat_07_08_2016) in DTC1 and SU-CCS-1 siCTL vs siEA samples, as suggested. As shown in **RtR_Fig1d**, siCTL and siEA conditions show similar signal intensities across the genome, and therefore we feel confident that the biological differences identified at the fusion protein binding sites are not linked to variations in ChIP-seq quality.

For Supplementary Fig. 2c, we acknowledge that the precise nature of the values plotted was unclear, and we apologize for this lack of clarity. The labels are ChIP-seq peak scores, and we have now edited the Methods and Figure legend accordingly to clarify this point. However, we respectfully disagree with the reviewer that this plot may suggest a “low correlation between SUCCS1 siEA and SUCCS1 siCTL genome wide, indicating possible problems in ChIP”. If this was the case, this issue would likely affect all H3K27ac signals genome wide in an unbiased manner, and therefore not causing the number of genome wide H3K27ac peaks to specifically increase in the siEA condition for both cell lines. Moreover, as pointed out by the reviewer, a potential reduced ChIP performance would be suspected to occur with the siEA samples, not siCTL samples, and in that case partly explain the observed reduction in the histone mark signals. On the contrary, we observe an increase in H3K27ac signals in siEA samples, that we don’t consider to be caused by ChIP problems. Moreover, our observation of increased enhancer activity upon EWSR1-ATF1 depletion do not rely uniquely on global H3K27ac signals as plotted in Supplementary Fig. 2c. We also show that the presence of “de novo” H3K27ac enhancers in siEA cells correlates with the emergence of *de novo* 3D loops in our Hi-ChIP data, and that the targets genes of these “de novo” enhancers are relevant for the lineage differentiation state of CCS cells.

10. “Overall, the two cell lines responded similarly to EWSR1-ATF1 depletion with a significant decrease in enhancer activity, as illustrated by the reduction in both H3K4me1 and H3K27ac signals at distal binding sites (Fig. 2c middle), as well as a reduction in H3K4me3 and H3K27ac signals at TSS sites (Extended data Fig. 2b).” Was this phenomenon specifically seen at the EWSR1-ATF1 binding sites? “genome wide H3K27ac signals showed both decreases and increases in EWSR1-ATF1-depleted DTC1 and SU-CCS-1 tumor cells (Extended data Fig. 2c), attesting to profound changes in their global epigenetic landscape.” How much could this be due to variation between replicates?

Response: Yes, we observed a decrease in enhancer activity specifically at EWSR1-ATF1 binding sites (Fig. 2c, Supplementary Fig. 2b), as the global H3K27ac signals were found to be increased in siEA samples (Supplementary Fig. 2c). The reviewer is also asking about the possibility that these changes may be due to variation between replicates. Although we are not entirely sure which replicates the reviewer is referring to, the ChIP-seq experiments were performed on one replicate of the siEA and siCTL conditions for each cell line. Despite the presence of some differences between the two cell lines, which are evident from the figures, the overall response we observed upon EWSR1-ATF1 depletion for both lines is very similar.

11. "The observed decrease in the ATAC signal also suggests that EWSR1-ATF1 may be involved in generating de novo distal regulatory elements through pioneering properties similar to the ones identified for EWSR1-FLI1 in Ews2." How do the authors deduct the pioneering function from these data?

Response: To the best of our knowledge, one of the key properties of pioneer TFs is their ability to directly bind nucleosomes at condensed chromatin regions and subsequently actively contribute to chromatin opening⁶. ATAC-seq is one of the most recognized and utilized technologies to assess chromatin accessibility, given that nucleosomal chromatin remains inaccessible to Tn5 Transposase activity⁷. ATAC-seq therefore provides a powerful mean to study the effect of pioneer TFs on chromatin accessibility. It follows that the reduction in ATAC-seq signal we observed in siEA samples indicates that upon EWSR1-ATF1 depletion the chromatin at the fusion protein binding sites has become less accessible. These results strongly suggest that EWSR1-ATF1 is involved in increasing chromatin accessibility. However, additional functional evidences also support the pioneer activity of EWSR1-ATF1:

- The low DNA accessibility profile of EWSR1-ATF1 bound sites in other cell types.
- The re-targeting of wt ATF1 to more distal sites.
- The interaction of EWSR1-ATF1 with BRG1, a core component of the SWI/SNF chromatin remodeling complex.
- The de novo increases in ATAC-seq and H3K27ac signals in hpMSCs upon EWSR1-ATF1 expression.
- The low accessibility of de novo and No ATF1 distal sites in other cell types.

"Wt ATF1 was found to be present at the majority of the EWSR1-ATF1 binding sites in both cell lines (62.5% in DTC1, 75% in SU-CCS-1, Fig. 3a), and confirmed in the primary CCS1 tumor, although at a lower percentage of EWSR1-ATF1 sites (38%, Fig. 3a), a difference that may be related to the technical challenge of profiling TFs in frozen tumor samples." Here the authors correctly acknowledge that ChIP quality issues may affect the results.

We once again we agree with the reviewer about the importance of ChIP-seq quality, and based on the analyses provided here we feel confident in our data.

12. "Consistent with our previous results, the binding pattern of EWSR1-ATF1 remained predominantly restricted to distal genomic regions (7% of TSSs), whereas the YS37 mutant displayed a significant increase in promoter binding (56% of TSS), close to the binding pattern of wt ATF1 (60% of TSS) (Fig. 4e)." Genome-wide correlation between TFs' ChIP-seq profiles would be much more informative.

Response: The reviewer proposes that a genome-wide correlation between TFs' ChIP-seq profiles would

be much more informative. We are not sure how a genome wide correlation would be more informative, as this experiment was done specifically to investigate whether mutating the EWSR1 prion-like domain would affect the genomic distribution of the fusion protein binding sites. Nevertheless, we have evaluated the genome wide Pearson correlation (**RtR_Fig2**) of the V5 ChIP-seq and the V5 signal at all TSSs (**RtR_Fig1d**), showing that the EWSR1(Y537)-ATF1 mutant correlates better with wt ATF1 than with EWSR1-ATF1. The EWSR1(Y537)-ATF1 and wt ATF1 also show stronger signal at all TSSs, as compared to EWSR1-ATF1. We believe that these observations support our results describing a marked change in binding site distribution for the EWSR1(Y537)-ATF1 mutant.

13. "These observations support the notion that CCS may originate from precursor cells along the neural crest lineage, and suggest that the fusion protein may induce a cellular reprogramming toward a more undifferentiated state by retargeting the lineage-specifying TFAP2A, SOX10 and MITF TFs from their original binding sites." The authors have profiled the TFAP2A binding sites by ChIP-seq in DTC1 and SUCCS-1 cells so it would be useful to see the change in the binding sites upon EWSR1-ATF1 removal to confirm this suggestion

Response: We fully agree this is an important experiment, and following the reviewer suggestion we have now profiled the TFAP2A, SOX10 and MITF TFs in SUCCS1 siEA vs siCTL cells. Our new analyses show that TFAP2A, SOX10 and MITF chromatin occupancy at the EWSR1-ATF1 binding sites decreases in siEA cells (Supplementary Fig. 2d), suggesting that their presence at these regulatory elements is, at least in part, EWSR1-ATF1-dependent.

14. "that EWSR1-ATF1 may display similar neomorphic pioneer properties that enable tumor initiation in CCS." Are these fusions strongly clonal in CCS (like they should be if tumor initiating)?

Response: Since EWSR1-ATF1 is the (hitherto identified) only recurrent genomic aberration in the vast majority of CCS tumors, and given that two mouse models of CCS have shown EWSR1-ATF1 expression to be sufficient to initiate tumor development, it is reasonable to think that EWSR1-ATF1 is an initiating and clonal, event in CCS development. Our results suggesting a pioneer function of EWSR1-ATF1 further supports the capability of this fusion protein to initiate tumor development. However, we are not aware of any study specifically investigating the clonal development of CCS tumor cells.

15. "Thus, the collaborative TF network identified in this study may serve EWSR1-ATF1 to induce the oncogenic reprogramming required for tumor development." The collaborative TF network proposed makes sense and it is conceivable that the tumor needs to work with factors present in the cell of origin; where there any other factors enriched?

Response: We have extended our analysis of collaborative TFs to also include SOX10 and MITF. SOX motifs were also enriched at EWSR1-ATF1 binding sites (Fig. 1d), and the rationale for profiling MITF is the

previously proposed involvement of MITF in CCS⁸ and the described collaborative function of TFAP2A, SOX10 and MITF in normal melanocytes and melanoma tumor cells⁹.

16. "CPM-normalized BigWig files were generated using bamCoverage" Are the reads extended to fragment size? They should be for visualization. The tools' documentation also says: By default, the read length is not extended. This is the preferred setting for spliced-read data like RNA-seq, where one usually wants to rely on the detected read locations only. A read extension would neglect potential splice sites in the unmapped part of the fragment. Other data, e.g. Chip-seq, where fragments are known to map contiguously, should be processed with read extension (--extendReads [INTEGER]).

Response: We apologize with the reviewer for the lack of clarity in describing this analysis in the Methods, section. The reads for all ChIP-seq experiments were extended to 300 bp, the mean fragment length. We have added this detail to the Methods of our revised manuscript.

References

- 1 Huang, G. L. *et al.* Activating transcription factor 1 is a prognostic marker of colorectal cancer. *Asian Pac J Cancer Prev* **13**, 1053-1057, doi:10.7314/apjcp.2012.13.3.1053 (2012).
- 2 Su, B. *et al.* Stage-associated dynamic activity profile of transcription factors in nasopharyngeal carcinoma progression based on protein/DNA array analysis. *OMICS* **15**, 49-60, doi:10.1089/omi.2010.0055 (2011).
- 3 Huang, G. L. *et al.* The protein level and transcription activity of activating transcription factor 1 is regulated by prolyl isomerase Pin1 in nasopharyngeal carcinoma progression. *Cell Death Dis* **7**, e2571, doi:10.1038/cddis.2016.349 (2016).
- 4 Zheng, D. *et al.* Cyclin-dependent kinase 3-mediated activating transcription factor 1 phosphorylation enhances cell transformation. *Cancer Res* **68**, 7650-7660, doi:10.1158/0008-5472.CAN-08-1137 (2008).
- 5 Ashoor, H., Kleftogiannis, D., Radovanovic, A. & Bajic, V. B. DENdb: database of integrated human enhancers. *Database (Oxford)* **2015**, doi:10.1093/database/bav085 (2015).
- 6 Zaret, K. S. Pioneer Transcription Factors Initiating Gene Network Changes. *Annu Rev Genet* **54**, 367-385, doi:10.1146/annurev-genet-030220-015007 (2020).
- 7 Buenrostro, J. D., Giresi, P. G., Zaba, L. C., Chang, H. Y. & Greenleaf, W. J. Transposition of native chromatin for fast and sensitive epigenomic profiling of open chromatin, DNA-binding proteins and nucleosome position. *Nat Methods* **10**, 1213-1218, doi:10.1038/nmeth.2688 (2013).
- 8 Davis, I. J. *et al.* Oncogenic MITF dysregulation in clear cell sarcoma: defining the MIT family of human cancers. *Cancer Cell* **9**, 473-484, doi:10.1016/j.ccr.2006.04.021 (2006).
- 9 Laurette, P. *et al.* Transcription factor MITF and remodeler BRG1 define chromatin organisation at regulatory elements in melanoma cells. *Elife* **4**, doi:10.7554/eLife.06857 (2015).

a

b

ChIP signal at ATF1 C /No EWSR1 N peaks

c

wt ATF1 expression in a selection of CCLE cell lines, compared to CCS

d

ChIP signal at all TSSs

Pearson correlation of ChIP replicates and samples

ATF1 C

EWSR1 N

ATF1 N

H3K4me3

H3K27ac

H3K4me1

V5 (YS37 mutant experiment)

a

ChIP signal at random genomic regions

b

ChIP signal at random genomic regions

ChIP signal at all TSSs

Reviewers' Comments:

Reviewer #1:

Remarks to the Author:

In the revised version of the manuscript, all my concerns are sufficiently addressed. I would suggest that the manuscript is acceptable for publication.

Reviewer #2:

Remarks to the Author:

The revised manuscript by Möller et al is complemented with new data and explaining text sequences that well answer and clarify the question marks and comments from this referee. The manuscript, in its present form, is a very impressive and important contribution to our understanding of the EWS-ATF1 fusion oncogene and its molecular action mechanisms. Moreover, it also expands our insight into oncogenic mechanisms of the whole group of more than 20 tumor entities caused by the same family of fusion oncogenes. The manuscript, in its revised form, is therefore recommended for publication in Nature communications!

Reviewer #3:

Remarks to the Author:

Möller et al. have revised their original submission reporting that EWSR1-ATF1 dependent 3D connectivity regulates oncogenic and differentiation programs in Clear Cell Sarcoma. The authors have addressed many of the points raised. The quality of ChIP-seq remains important for the contribution and to that end reporting the FRiP scores for the experiments would further improve the work.

Two specific items one would still like to look into:

1) "The reviewer raises the concern about ChIP-seq quality differences between the siCTL and siEA conditions, as a potential reason explaining the lower histone mark signal identified in siEA CCS cells. We fully agree that this is a valid question, and to address this point we generated heatmaps of H3K27ac and ATAC signals at all TSSs genome wide (ucsc_RefFlat_07_08_2016) in DTC1 and SU-CCS-1 siCTL vs siEA samples, as suggested. As shown in RtR_Fig1d, siCTL and siEA conditions show similar signal intensities across the genome, and therefore we feel confident that the biological differences identified at the fusion protein binding sites are not linked to variations in ChIP-seq quality."

Could the authors show the data presented in RtR_Fig1d as a composite plot next to Suppl Fig 2b instead of a heatmap (that is not very informative quantitatively)? This would allow the reader to evaluate better the statement in the manuscript text "Overall, the two cell lines responded similarly to EWSR1-ATF1 depletion with a significant decrease in enhancer activity, as illustrated by the reduction in both H3K4me1 and H3K27ac signals at distal binding sites (Fig. 2c middle), as well as a reduction in H3K4me3 and H3K27ac signals at TSS sites (Supplementary Fig. 2b)."

2) "For Supplementary Fig. 2c, we acknowledge that the precise nature of the values plotted was unclear, and we apologize for this lack of clarity. The labels are ChIP-seq peak scores, and we have now edited the Methods and Figure legend accordingly to clarify this point. However, we respectfully disagree with the reviewer that this plot may suggest a "low correlation between SUCCS1 siEA and SUCCS1 siCTL genome wide, indicating possible problems in ChIP". If this was the case, this issue would likely affect all H3K27ac signals genome wide in an unbiased manner, and therefore not causing the number of genome wide H3K27ac peaks to specifically increase in the siEA condition for both cell lines. Moreover, as pointed out by the reviewer, a potential reduced ChIP performance would be

suspected to occur with the siEA samples, not siCTL samples, and in that case partly explain the observed reduction in the histone mark signals. On the contrary, we observe an increase in H3K27ac signals in siEA samples, that we don't consider to be caused by ChIP problems. Moreover, our observation of increased enhancer activity upon EWSR1- ATF1 depletion do not rely uniquely on global H3K27ac signals as plotted in Supplementary Fig. 2c. We also show that the presence of "de novo" H3K27ac enhancers in siEA cells correlates with the emergence of de novo 3D loops in our Hi-ChIP data, and that the targets genes of these "de novo" enhancers are relevant for the lineage differentiation state of CCS cells."

Here there seems to be some confusion in the response. In the manuscript text the authors say that enhancer activity is decreased in siEA cells but here the message is quite the opposite.

Response to REVIEWER'S COMMENTS

We would like to thank the reviewers for their positive feedback on the improved quality of our revised manuscript.

We have now addressed the remaining remarks raised by Reviewer #3, and therefore hope that this new version of our study will be suitable for publication in Nature Communications.

Reviewer #1 (Remarks to the Author):

In the revised version of the manuscript, all my concerns are sufficiently addressed. I would suggest that the manuscript is acceptable for publication.

We would like to thank the reviewer for helping us improving the quality and impact of our work.

Reviewer#2 (Remarks to the Author):

The revised manuscript by Möller et al is complemented with new data and explaining text sequences that well answer and clarify the question marks and comments from this referee. The manuscript, in its present form, is a very impressive and important contribution to our understanding of the EWS-ATF1 fusion oncogene and its molecular action mechanisms. Moreover, it also expands our insight into oncogenic mechanisms of the whole group of more than 20 tumor entities caused by the same family of fusion oncogenes. The manuscript, in its revised form, is therefore recommended for publication in Nature communications!

We thank the reviewer for agreeing on the novelty and impact of this work, and for pointing us to key experiments that helped improving its quality.

Reviewer #3 (Remarks to the Author):

Möller et al. have revised their original submission reporting that EWSR1-ATF1 dependent 3D connectivity regulates oncogenic and differentiation programs in Clear Cell Sarcoma. The authors have addressed many of the points raised. The quality of CHIP-seq remains important for the contribution and to that end reporting the FRiP scores for the experiments would further improve the work.

Two specific items one would still like to look into:

1) "The reviewer raises the concern about CHIP-seq quality differences between the siCTL and siEA conditions, as a potential reason explaining the lower histone mark signal identified in siEA CCS cells. We

fully agree that this is a valid question, and to address this point we generated heatmaps of H3K27ac and ATAC signals at all TSSs genome wide (ucsc_RefFlat_07_08_2016) in DTC1 and SU-CCS-1 siCTL vs siEA samples, as suggested. As shown in RtR_Fig1d, siCTL and siEA conditions show similar signal intensities across the genome, and therefore we feel confident that the biological differences identified at the fusion protein binding sites are not linked to variations in CHIP-seq quality.”

Could the authors show the data presented in RtR_Fig1d as a composite plot next to Suppl Fig 2b instead of a heatmap (that is not very informative quantitatively)? This would allow the reader to evaluate better the statement in the manuscript text "Overall, the two cell lines responded similarly to EWSR1-ATF1 depletion with a significant decrease in enhancer activity, as illustrated by the reduction in both H3K4me1 and H3K27ac signals at distal binding sites (Fig. 2c middle), as well as a reduction in H3K4me3 and H3K27ac signals at TSS sites (Supplementary Fig. 2b)."

Response: we agree with the reviewer that while heatmaps provide a better image of signal distributions genome wide, composite plots give a more precise representation of signal changes between different conditions.

Following the reviewer suggestion, we have now added composite plots for the data presented in RtR_Fig1d to our new Suppl Fig 2b. To generate this new plots we compared changes in H3K27ac and ATAC signal intensities upon EWSR1-ATF1 depletion between our 374 TSSs bound by EWS-ATF1 and a similar number of active TSSs that are not bound by the fusion protein. These new data further support the notion that the changes in chromatin activation and accessibility observed at EWSR1-ATF1 binding sites in siEA CCS cells are not the result of decreased CHIP-seq and ATAC-seq quality.

2) “For Supplementary Fig. 2c, we acknowledge that the precise nature of the values plotted was unclear, and we apologize for this lack of clarity. The labels are CHIP-seq peak scores, and we have now edited the Methods and Figure legend accordingly to clarify this point. However, we respectfully disagree with the reviewer that this plot may suggest a “low correlation between SUCCS1 siEA and SUCCS1 siCTL genome wide, indicating possible problems in CHIP”. If this was the case, this issue would likely affect all H3K27ac signals genome wide in an unbiased manner, and therefore not causing the number of genome wide H3K27ac peaks to specifically increase in the siEA condition for both cell lines. Moreover, as pointed out by the reviewer, a potential reduced CHIP performance would be suspected to occur with the siEA samples, not siCTL samples, and in that case partly explain the observed reduction in the histone mark signals. On the contrary, we observe an increase in H3K27ac signals in siEA samples, that we don’t consider to be caused by CHIP problems. Moreover, our observation of increased enhancer activity upon EWSR1- ATF1 depletion do not rely uniquely on global H3K27ac signals as plotted in Supplementary Fig. 2c. We also show that the presence of “de novo” H3K27ac enhancers in siEA cells correlates with the emergence of de novo 3D loops in our Hi-ChIP data, and that the targets genes of these “de novo” enhancers are relevant for the lineage differentiation state of CCS cells.”

Here there seems to be some confusion in the response. In the manuscript text the authors say that enhancer activity is decreased in siEA cells but here the message is quite the opposite.

Response: we apologize if our previous reply was unclear on this particular point. We would like to emphasize that in our study we examined two distinct sets of H3K27ac sites:

1. The 2385 genomic sites that correspond to the fusion protein binding regions: these sites show decreases in H3K27ac signal in siEA-treated cells (Fig. 2c, Supplementary Fig. 2b), consistent with the chromatin activation function of EWSR1-ATF1.
2. The remaining genome-wide H3K27ac sites that are not associated with EWSR1-ATF1 binding, and which display both signal increases and decreases in siEA-treated cells (Supplementary Fig. 2c). These bi-directional changes testify against a general decrease in H3K27ac signal in siEA cells, which could have been potentially associated with a lower ChIP-seq quality in these cells. Moreover, among the increased H3K27ac regions in siEA cells, we also identified a set of “*de novo*” sites defined as H3K27ac peaks showing less than 5 reads in siCTL cells, and increasing at least four-fold in siEA cells (Fig. 7a). The direct target genes of these “*de novo*” enhancers were then identified by Hi-ChIP data (Fig. 7b), and shown to be involved into terminal melanocytic differentiation (Fig. 7c-f).

In aggregate, our study identifies the direct binding sites of the fusion protein, which show decrease in H3K27ac signal upon EWSR1-ATF1 depletion, as well as a set of H3K27ac sites which emerge “*de novo*” in siEA-treated CCS cells, and are associated with the terminal differentiation induced by the loss of the fusion protein. Taken together, these observations illustrate that the changes in H3K27ac signal that follow EWSR1-ATF1 depletion are not the result of decreased ChIP-seq quality in these cells.

We have now edited the figures and text accordingly and we hope that the reviewer will find our answers clear and satisfying.

Reviewers' Comments:

Reviewer #3:

Remarks to the Author:

Many thanks for addressing the issues. I have no further comments.